
# A meta-analysis of groundwater contamination by nitrates at the
# African scale
Issoufou Ouedraogo[1] and Marnik Vanclooster[1]
[1]Earth and Life Institute, Université catholique de Louvain, Croix du Sud 2, Box 2, B-1348 Louvain-la-Neuve, Belgium.
*Correspondence to*:  I. Ouedraogo (ouediss6@yahoo.fr)
**Abstract.** Contamination of groundwater with nitrate poses a major health risk to millions of people around Africa.
Assessing the space-time distribution of this contamination, as well as understanding the factors that explain this
contamination is important to manage sustainable drinking water at the regional scale. This study aims assessing the
variables that contribute to nitrate pollution in groundwater at the pan African scale by statistical modeling. We compiled
a literature database of nitrate concentration in groundwater (around 250 studies) and combined it with digital maps of
physical attributes such as soil, geology, climate, hydrogeology and anthropogenic data for statistical model
development. The maximum, medium and minimum observed nitrate concentrations were analysed. In total, 13
explanatory variables were screened to explain observed nitrate pollution in groundwater. For the mean nitrate
concentration, 4 variables are retained in the statistical explanatory model: (1) Depth to groundwater (shallow
groundwater, typically <50m); (2) Recharge rate; (3) Aquifer type; and (4) Population density. The former three
variables represent intrinsic vulnerability of groundwater systems towards pollution, while the latter variable is a proxy
for anthropogenic pollution pressure. The model explains 65% of the variation of mean nitrate contamination in
groundwater at the pan Africa scale. Using the same proxy information, we could develop a statistical model for the
maximum nitrate concentrations that explains 42% of the nitrate variation. For the maximum concentrations, other
environmental attributes such as soil type, slope, rainfall, climate class and region type improves the prediction of
maximum nitrate concentrations at the pan African scale. As to minimal nitrate concentrations, in the absence of normal
distribution assumptions of the dataset, we do not develop a statistical model for these data. The data based statistical
model presented here represents an important step toward developing tools that will allow us to accurately predict nitrate
distribution at the African scale and thus may support groundwater monitoring and water management that aims
protecting groundwater systems. Yet they should be further refined and validated when more detailed and harmonized
data becomes available and/or combined with more conceptual descriptions of the fate of nutrients in the hydrosystem.





**1 Introduction**
Nitrate contamination of groundwater is a common problem in many parts of the world. Elevated nitrate concentrations
in drinking water can cause methemoglobinemia in infants and stomach cancer in adults (Yang et al., 1998; Knobeloch
et al., 2000; Hall et al., 2001). As such, the World Health Organization (WHO) has established a maximum contaminant
level (MCL) of 50 mg/L $NO_3$ (WHO, 2004). Nitrate in groundwater is generally from anthropogenic origin and
associated with leaching of nitrogen from agriculture plots or from waste and sewage sanitation systems. The heavy use
of nitrogenous fertilizers in cropping system is the largest contributor to anthropogenic nitrogen in groundwater
worldwide (Suthar et al., 2009). In particular, shallow aquifers in agricultural fields are highly vulnerable to nitrate
contamination (Böhlke, 2002; Kyoung-Ho et al, 2009). According to Spalding and Exner (1993), nitrate may be the
most widespread contaminant of groundwater.
In Africa, groundwater is recognized as playing a very important role in the development agenda. However, according
to Xu and Usher, (2006), degradation of groundwater is the most serious water resources problem in Africa. The two
main threats are overexploitation and contamination (MacDonald et al., 2013). Indeed, based on a review of 29 papers
from 16 countries, Xu and Usher (2006), have identified major groundwater pollution issues in Africa, considering the
following order of importance as follows: (1) nitrate pollution, (2) pathogenic agents, (3) organic pollution, (4)
salinization, and (5) acid mine drainage. These authors have identified that the major sources of groundwater
contamination are related to on-site sanitation, to the presence of solid waste dumpsites, including household waste pits,
to infiltration of surface water, to agricultural activities, to the presence of petrol service stations (underground storage
tanks), and to the mismanagement of wellfields. Nitrate contamination of groundwater is a problem that commonly
occurs in Africa, as illustrated in the studies for Algeria (Rouabhia et al., 2010; Messameh et al., 2014), Tunisia (Hamza
et al., 2007; Anane et al., 2014), Morocco (Bricha et al., 2007; Fetouani et al., 2008; Benabbou et al., 2014), Senegal
(Sall and Vanclooster, 2009; Diédhiou et al., 2012), Ivory Coast (Loko et al., 2013; Eblin et al., 2014), Ghana (Tay and
Kortatsi, 2008; Fianko et al., 2009 ), Nigeria (Wakida and Lerner, 2005; Akoteyon and Soladoye, 2011; Obinna et al.,
2014), South Africa (Maherry et al., 2009; Musekiwa and Majola, 2013), Ethiopia (BGS, 2001; Bonetto et al., 2005)
and Zambia (Wakida and Lerner, 2005). Several of these studies showed that pollution from anthropogenic activities
is the main source of high and variable nitrate levels. For example, Comte et al., (2012) illustrates that the groundwater
situated in the Quaternary sandy aquifer of the peninsula of Dakar is under strong anthropogenic pressure from the city
of Dakar, resulting in important nitrate loadings. Such contamination problems are often retrieved in many metropoles
in Africa. Notwithstanding the availability of all these studies at the local, regional or country level, no comprehensive
and synthetic study of nitrate contamination of groundwater at the scale of the African continent has been presented in
the literature. Assessing large scale groundwater contamination with nitrates is important for the planning of the large
scale groundwater exploitation programs and for designing transboundary water management policies. It yields also
important baseline information for monitoring progress in the implementation of the UN SDGs for water. It increase
awareness of citizens, international agencies and authorities (e.g., FAO, UNEP, and OECD, Water Sanitation for Africa
(WSA)) on the environmental factors likely to be significant to groundwater contamination. However, making an
appropriate pan-African synthesis of nitrate contamination of groundwater remains a scientific and technical challenge,
given the non-homogeneity of the nitrate monitoring programmes and the absence of administrative and institutional
capacity to collect and diffuse the data at the pan African scale. A concept that partially helps solving this urgent data
management problem is the concept of groundwater vulnerability. Groundwater vulnerability for nitrate contamination



is an expression of the likelihood that a given groundwater body will be negatively affected by nitrate contamination.
Given that the vulnerability is a likelihood, it is only an expression of the potential degradation of groundwater and
hence a proxy of groundwater contamination by nitrates. Groundwater vulnerability can be assessed based on available
generic data. It does therefore not depend on a strong and operational pan African groundwater quality monitoring
capacity. In this paper we propose and implement a methodology for assessing vulnerability of groundwater
contamination by nitrates at the pan African scale. We further consider nitrate in this study as a proxy for overall
groundwater pollution, which is consistent with the view of the US EPA (EPA, 1996).

In general, there are three categories of models for the assessment of groundwater vulnerability: (1) index methods or
subjective rating methods, (2) statistical methods and (3) process based modelling methods. Index-and-overlay methods
are one set of subjective rating methods that utilize the intersection of regional attributes with the qualitative
interpretation of data by indexing parameters and assigning a weighting scheme. The most widely used index method
is DRASTIC (Aller et al., 1985). Unfortunately, index methods are based on subjective rating methods (Focazio et al.,
2002) and should preferably be calibrated using measured proxies of vulnerability (Kihumba et al., 2015; Ouedraogo et
al., 2015). When a groundwater monitoring dataset is available, formal statistical methods can be used to integrate
groundwater contamination data directly in the vulnerability assessment. Finally, process-based methods refer to
approaches that explicitly simulate the physical, chemical and biological processes that affect contaminant behaviour in
the environment. They comprise the use of deterministic or stochastic process-simulation models, eventually linked to
physically based field observations (e.g., Coplen et al., 2000). Physically process-based methods are typically applied
at small scales, mostly to define well protection zones, rather than to assess groundwater vulnerability at broader scales
(Frind et al., 2006). A well-known example is the use of a physical based groundwater model (e.g. MODFLOW,
Harbaugh et al., 2000) that solve the governing equations of groundwater flow and solute transport. Such models have
explicit time steps and are often used to determine the time scales of contaminant transport to wells and streams, in
addition to the effects of pumping. However, they also have many parameters that require estimation. In this paper, we
use statistical models to assess vulnerability of groundwater systems towards nitrate pollution.

Formal statistical methods have often be deployed to assess vulnerability of groundwater at national and regional scales.
They are also often used to discriminate contaminant sources and to identify factors contributing to contamination
(Kolpin, 1997; Nolan and Hitt, 2006). Many authors used multiple linear regression (MLR) techniques. For example,
Bauder et al., (1993) investigated the major controlling factors for nitrate contamination of groundwater in agricultural
areas using MLR of land uses, climate, soil characteristics, and cultivations types. MLR was also used to relate pesticide
concentrations in groundwater to the age of the well, land use around the well, and the distance to the closest possible
source of pesticide contamination (Steichen, et al., 1988). Boy-Roura et al., (2013) used MLR to assess nitrate pollution
in the Osona region (NE Spain). Amini et al., (2008a) and Amini et al., (2008b) used MLR and Adaptive Neuro-Fuzzy
Inference System (ANFIS), a general non-linear regression technique, to study the global geogenic fluoride
contamination in groundwater and the global geogenic arsenic contamination in groundwater respectively. MLR has
the strong advantage that regression coefficients can directly be interpreted in terms of importance of explaining factors.
Many studies linking nitrate occurrence in groundwater to spatial variables have employed logistic regression (Hosmer
and Lameshow, 1989; Eckardt and Stackkelberg, 1995; Tesoriero and Voss, 1997; Gardner and Vogel, 2005; Winkel
et al., 2008; and Mair and El-kadi, 2013). According to Kleinbaum. (1994), MLR is conceptually similar to logistic
regression. Other authors have used more sophisticated approaches such as Bayesian methods (Worrall and Besien,





2005; Mattern et al., 2012) and, more recently, classification and regression tree modelling approaches (Burow et al.,
2010; Mattern et al., 2012). Yet, according to our knowledge, a statistical model of groundwater nitrate contamination
at the pan African scale does not exist yet.
In the present study, we used MLR techniques to assess the vulnerability of nitrate groundwater pollution at the pan
African scale. To this end, we compiled a pan African groundwater pollution data base from the literature and combined
it with environmental attributes inferred from a generic data basis. The generic data basis was developed in a former
study to assess vulnerability using the DRASTIC index method (Ouedraogo et al., 2016). MLR models were
subsequently identified to explain quantitatively the logtransformed observed nitrate contamination in terms of generic
environmental attributes and finally, the regression models were interpreted in terms of characteristics of contaminants
sources and hydrogeology of the African continent.

## 122  2 Study area


We studied the vulnerability of groundwater systems for nitrate contamination at the scale of the African continent.
Groundwater is Africa's most precious natural resource, providing reliable water supplies to at least a third of the
continent's population (MacDonald, 2010). However, the African continent is not blessed by a large quantity of
groundwater resources, because it is the World's second-driest continent after Australia and water resources are limited.
MacDonald et al., (2012) have estimated the volume of groundwater resource in Africa at 0.66 million $km^3$.

Africa has a vast array of drainage networks, the most important ones are the Nile River, which drains northeast and
empties into the Mediterranean Sea. The Congo River drains much of central Africa and empties into the Atlantic Ocean.
The Niger River is the principal river of western Africa; it is the third-longest river after the Nile and the Congo River
and empties into the Atlantic Ocean. Southern Africa is drained by the Zambezi River. Lake Chad constitutes one of
the largest inland drainage areas of the continent. Other major lakes located in the east of Africa include Lake
Tanganyika and Lake Victoria.

The elevation of Africa varies from below sea level to 5825 m above sea level. The average elevation is approximately
651 m (Ateawung, 2010). The geology of the African continent contains 13 lithological classes (Fig.1) with varying
coverages: evaporites (0.6%), metamorphic rocks (27.6%), acid plutonic rocks (1.1%), basic plutonic rocks (0.2%),
intermediate plutonic rocks (0.1%), carbonates sedimentary rocks (9.4%), mixed sedimentary rocks (6.4%), siliciclastic
sedimentary rocks (16.4%), unconsolidated sediments (35.1%), acid volcanic rocks (0.1%), basic volcanic rocks (3.3%),
intermediate volcanic rocks (0.6%) and water bodies (0.9%) (Hartmann and Moosdorf, 2012). The lithology describes
the geochemical, mineralogical and physical properties of rocks.

## 145  3 Data and methods

### 146  3.1 Nitrate contamination data




For a large part of Africa there is very little, or no systematic monitoring of groundwater. In the absence of data
systematic monitoring program, we compiled nitrate pollution data at the pan African scale from different literature
sources. We considered approximately 250 published papers on nitrate contamination of groundwater in Africa. We
consulted the web of sciences (Scopus™, Sciences Direct™, Google™, and Google Scholar™) and available books.
Fig. 2 shows the spatial distribution of the considered field studies. Table 1 outlines criteria used in the web search.

**3.2 Data quality evaluation**

We used the following additional criteria to select the study:
i.    the publication should explicitly report on nitrate concentrations in groundwater; and
ii.   the publication should be published after 1999.
Also, when many articles have been published for the same field site, we used only the most recent study. We excluded
older studies before 1999, since the intensity of human activities is expected to be significantly different after 1999. We
eliminated thirty-seven articles because no quantitative data on nitrate concentration were reported. For the considered
data set, 206 studies report on the maximum concentration of nitrate, 187 studies on minimum concentration of nitrate,
and 94 studies on the mean concentration of nitrate. Out of the 94 datasets for which mean values were reported, 12
field sites have nitrate concentration smaller than 1 mg/L. We present the locations and references of the considered
field studies in Table 2. In case spatial coordinates were not reported in the selected paper, we allocated the coordinates
of the field study in Google Earth using the www.gps-coordintes.net and www.mapcoordinates.net applications. As an
example, we present in Fig. 3 the identified locations and reported maximum nitrate values of the selected studies. The
absence of exact spatial coordinates in many studies will therefore generate a positioning error in the analysis. However,
given the extent of the study, i.e. the African continent, we consider that this positioning error will not have significant
effects on the overall results.

**3.3 Determination of spatial explanatory variables**

Table 3, list the environmental attributes and data sources that we considered for explaining the observed nitrate
contamination. These variables represent both anthropogenic and natural factors and were derived from multiple sources
of information. The attributes are related to recharge, geology, hydrogeology, soil texture, land use, topography and
pollution pressure and were partially inspired from the DRASTIC vulnerability mapping approach. We compiled all
explanatory variables in a common GIS environment (ArcGIS 10.3™), using a common projection and resolution (15
km x 15 km) at the 1:60.000.000 scale. This spatial resolution was chosen because, we have considered that she was a
reasonable compromise between different resolutions of the different datasets, computing constraints and regional
extent. Indeed, this grid cell dimension has been used to map the vulnerability and risk pollution maps at the African
scale (Ouedraogo et al., 2016). Generic variables at the grid scale were extracted to build our explanatory variables in
this study. Most of these variables were categorical, but some were continuous.

Groundwater recharge is considered as primary explaining variable because recharge is the primary vehicle by which a
contaminant is transported from the ground surface to groundwater. Groundwater recharge to an unconfined aquifer is





a function of precipitation, runoff, and evapotranspiration. The latter is related to vegetation and/or soil type.
Groundwater recharge to a confined aquifer is generally more complex, as consideration must be given to the location
of the recharge zone and the influence of any confining layers, vertical gradients, and groundwater pumping (Todd and
Kennedy, 2010). In this study, we derived the African recharge map from the global-scale groundwater recharge model
of Döll et al. (2008). We also considered independent climate data as alternative proxies of recharge.  Hence, we
considered the climate and region type data class as defined by Trambauer et al. (2014). We also considered the rainfall
map as generated from the UNEP/FAO World and Africa GIS Data Base. The spatial resolution of this latter dataset is
approximately 3.7 kilometers.
Subsequently we selected a set of environmental attributes related to aquifer type, groundwater position and the substrate
that protects the aquifer.  The depth to groundwater represents the distance that a contaminant must travel through the
unsaturated zone before reaching the water table or to the first screen. We mapped the depth to water based on the data
presented by Bonsor et al. (2011). The slope of the land surface is important with respect to groundwater vulnerability,
because it determines the potential of a contaminant to infiltrate into the groundwater or be transported horizontally as
runoff.  We inferred the slope from the 90 meter Shuttle Radar Topography Mission (SRTM90) topographic map, using
the Spatial Analyst software of ArcGIS10.2™. We derived the aquifer type and the impact of vadose zone material from
the high resolution global lithological database (GliM) of Hartmann and Moosdorf (2012). We determined aquifer type
and unsaturated lithological zone for each of the five hydro-lithological and lithological categories as defined by Gleeson
et al., (2014). These categories are: unconsolidated sediments, siliciclastic sediments, carbonate rocks, crystalline rocks,
and volcanic rocks (Gleeson et al., 2014). We constructed the soil type map from the 1 km resolution soil grid database
developed by Hengl et al. (2012). We determined the hydraulic conductivity of aquifers from the Global Hydrogeology
MaPS (GHYMPS) dataset (Gleeson et al., 2014). For the determination of the land use at the pan-African scale, we
used the high resolution land cover/land use map from the GlobCoverdataset (Defourny et al.,2014). There are twenty
two (22) classes of land cover that represents Africa in this dataset. We aggregated these 22 classes into 6 similar classes
(water bodies, bare area, grassland/shrubland, forest, urban, croplands) as represented in the Fig. 4 and then regrouped
them in 5 groups (water bodies, forest/bare area, grassland/shrubland, croplands, urban area).
Finally, we considered a set of variables related to possible pollution pressure. We considered the application of fertilizer
in the agricultural sector as a possible explanatory variable. We generated the nitrogen fertilizer application map from
the Potter and Ramankutty (2010) dataset. The values shown on this map represent an average application rate for all
crops over a 0.5° resolution grid cell. Following this study, the highest N fertilizer application rate (i.e. 220 kg / ha) is
found in Egypt's Nile Delta. We further considered population density as a proxy of pollution source.  We considered
the population density map for the year 2000, as produced by Nelson (2004).

### 3.4 Statistical model description

We used Multiple Linear Regression (MLR) as the statistical method for identifying the relationship between the
observed nitrate concentrations in groundwater and the set of independent variables given in Table 3. MLR is based on
least squares, which means that the model is fitted such that the sum of squares of differences of predicted and measured
values is minimized (Koklu et al., 2009; Helsel and Hirsh. (1992)). The MLR model is denoted as by Eq. (1):




$y_i = \beta_0 + \sum_{j=1}^{n} \beta_j x_{ij} + \varepsilon_i$        i=1, m,                                                                  (1)
where $y_i$ is the response variable at location i, $\beta_0$ is the intercept, $\beta_j$ are the slope coefficients of the explanatory
categorical or continuous  variables $x_{ij}$, n the number of variables and m is the  number of locations or wells (number
of studies here). $\varepsilon_i$ is the regression residual. In this study, the response variable is the logtransformed nitrate
concentration in groundwater. The logtransformation was need to stabilize the variance and to comply with the basic
hypothesis of MLR. The logtransformed nitrate concentration is a continuous monotonic increasing function; it is
therefore reasonable to accept that factors that contribute to the logtransformed nitrate load will also contribute to the
nitrate load. The explanatory variables were defined using a stepwise procedure, using the Akaike Information Criterion
(AIC) as test statistic (Helsel and Hirsch, 1992).  We evaluated model performance based on the significance level of
estimated coefficients, the coefficient of determination ($R^2$), the mean square error (MSE), the probability plots of model
residuals (PRES), the plots of predicted versus observed values and the Akaike Information Criterion (AIC). High values
of $R^2$ and low values of RMSE, PRES and AIC indicate a better performance of the model. To validate the model
obtained by the stepwise procedure, the standard regression diagnostics were assessed. To test the heteroscedasticity in
the model residuals, we use the Breusch-Pagan (BP) test by implementing with "lmtest" package. A Student statistic t
test was finally used to check the statistical significance (with p-values <0.10) of variables in the final model.  We
assessed tolerance to examine if multicollinearity exists between variables. In this study, we performed the statistical
analyses using the R version 3.1.1 (R Development Core team, 2015).
**4 Results**
**4.1 Normality of the dependent variable**
Prior to analysis, we carefully checked the data using descriptive statistics, such as boxplots and correlation analysis.
The observed nitrate concentrations through meta-analysis ranges from 0 mg/L to 4625 mg/L for all categories, i.e.
mean, maximum and minimum values of nitrate groundwater contamination. Descriptive statistics are summarized in
Table 4. The average mean nitrate concentration is 27.85 mg/L. The positive skewness of the mean nitrate concentration
data and the kurtosis suggest that the mean nitrate concentration is not normally distributed. In contrast, the lognormally
transformed mean nitrate concentration obeys normality, as demonstrated by means of the non-parametric Shapiro-Wilk
test (p-value=0.1432>0.05). The histogram of mean and logtransformed concentration is shown in the Fig. 5. We also
checked the minimum and maximum nitrate concentration for normality (results can be obtained from the authors upon
request).

**4.2 Correlation between nitrate in groundwater and explanatory variables**

Land Cover/Land Use is a principle factor, controlling groundwater contamination. The box plot distribution of
logtransformed mean nitrate concentration for different land use classes is presented in Fig. 6.  Groundwater in
agricultural and urban areas is clearly more susceptible to nitrate pollution as compared to forest/bare area land use.
Also water bodies are susceptible to nitrate contamination but this result is likely spurious since only two studies support
this category. We performed a similar analysis on the logtransformed maximum and minimum nitrate concentration.
The corresponding boxplots results can be obtained from the authors upon request. High values for logtransformed
maximum nitrate concentration are also found in urban and cropland areas. High values for logtransformed minimum





nitrate concentration are detected in croplands fields. All analyses confirm that the highest nitrate pollution is retrieved
in urban areas, immediately followed by agricultural areas.

In this study, the aquifer systems for Africa are divided into 5 categories based on the lithological formations. Fig. 7
shows the relation between mean logtransformed nitrate concentration and aquifer system type class. The carbonates
rocks, the unconsolidated sediments and the siliciclastic sedimentary rocks, represents respectively the first, the second
and the third class in terms of nitrate contamination. The crystalline rock and volcanic rock aquifer classes are less
contaminated. The high concentrations in the unconsolidated aquifer systems is a particular point of concern, since this
class is the most representative in terms of groundwater exploitation. The high concentrations in the carbonates rocks
and fractured basalt can be explained by their high vulnerability related to the presence of solution channels and
fractures.

The distribution of the logtransformed mean nitrate concentration data with depth is shown in Fig. 8. Apparently, no
clear relationship exist between depth to groundwater and nitrate contamination. The Pearson's correlation give a poor
correlation (r=0.004). However, the careful analysis of this figure shows clearly that shallower wells (7-25 m bgl and
25-50 m bgl) are associated with higher values of logtransformed mean nitrate concentration, in contrast to the low
values of logtransformed nitrate concentrations found in the deeper groundwater systems ( >250 m bgl).

The relationship between the logtransformed mean nitrate concentration and groundwater recharge can also be observed
in Fig. 8. This figure shows that nitrate concentration in the groundwater of shallow aquifers generally increases with
recharge, except for the very low recharge class (0-45 mm/year). The high nitrate observations for this latter low
recharge class may be due to irrigation water return that feed the groundwater and that is not integrated in the recharge
calculations. The analysis of Pearson's correlation between recharge and logtranformed mean nitrate give an r=-0.292.

The relation between the logtransformed mean nitrate concentration and the population density is given in Fig. 8. We
observe an increasing nitrate in groundwater related to increasing population. This explicit relationship between
population density and nitrate concentration has a Pearson's correlation of 0.632. This obviously confirms the
importance of studying the population as a potential polluting parameter and its relevant correlation to nitrate occurrence
in the groundwater at the pan African scale.

Nitrogen fertilizer contributes significantly to an increase in crop yields, but excess nitrogen fertilizer generally pollutes
groundwater (Green et al., 2005; Nolan et al., 2002). In the case of Africa, the impact of the nitrogen fertilizer application
rate on logtransformed mean nitrate concentration is illustrated in Fig. 8. Pearson's correlation give a low relation
(r=0.09). The analysis in this figure confirms that no clear relationship exist between fertilizer load and groundwater
nitrate contamination. This can be linked to the relatively low fertilizer use in Africa, as compared to other continents.
Indeed, most studies have nitrogen fertilizer dressings that are below 50 kg/ha. According the FAO (2012), Africa
accounts only for about 2.9 percent of the world fertilizer consumption in 2011.






We performed similar correlation analysis on the logtransformed maximum concentration and logtransformed minimum
concentration respectively. Details can be obtained from the authors upon request. Results of these analyses are coherent
with the results for logtransformed mean nitrate concentration.

**4.3 Development of the multi-variate statistical model**

We developed a set of multiple variable regression models for the logtransformed mean and maximum nitrate
concentration in terms of above mentioned explanatory variables. A positive regression coefficient indicates a positive
correlation between a significant explanatory variable and a target contaminant, while a negative coefficient suggests
an inverse or negative correlation. We retained only explanatory variables with p-values ≤ 0.1.

The best final model that explains the log transformed mean nitrate concentration includes only 4 explanatory variables:
(1) Depth to groundwater, (2) Recharge, (3) Aquifer type, and (4) Population density. Table 5 summarizes the results
of this linear regression model. This model can explain 65 percent of the logtransformed mean nitrate concentration
observations. The sign of the parameter coefficient indicates the direction of the relationship between independent and
dependent variable (Boy-Roura et al., 2013). The p-value expresses the «attained significance level" for that slope
coefficient, which is the significance level attained by the data (Helsel and Hirsch, 1992). The lower the p-value, the
more significant is the model parameter.

The regression analysis confirms the strong relationship between population density and logtransformed mean nitrate
concentration. As the p-value is far below 0.05, we are more than 95 % confident that the population density strongly
affects the nitrate occurrences in groundwater.

The aquifer medium is another important explanatory variable for logtransformed mean nitrate concentration. Three
categories of aquifer media are significantly explaining the dependent variables: carbonates rocks, crystalline and
unconsolidated sediments rocks. Indeed, the analyse of regression coefficients shows that the likelihood of nitrate
contamination decreases with the presence of unconsolidated sediments and crystalline rocks. Other aquifer types tested
include siliciclastic sedimentary rocks and volcanic rocks aquifers were found not statistically significant in the model.
However, the aquifer media type is an important variable to assess groundwater vulnerability and to bring information
about the hydrogeological system in the assessment. It allows differentiating the vulnerability in terms of aquifer
lithology. Variables such as hydraulic conductivity could be surrogates for aquifer media, because hydraulic
conductivity data were developed based on the lithological formation. Nevertheless they were not statistically significant
in the final model.

The third variable represents the depth to groundwater. The three first classes (0-7; 7-25 and 25-50 m bgl) of
groundwater depth are all statistically significant. The water table corresponding to the 0-7 m class has the strongest
statistical significance. The positive parameter coefficient indicates large contamination for shallow groundwater
depths. By analysing the table of coefficient, we observe that the largest groundwater depth class (100-250 m bgl) is not
statistically significant (p-value >0.05). We can conclude that the shallow groundwater systems in African scale are
most vulnerable to nitrate pollution.



The fourth variable included in the final model is the recharge. The recharge rate in the 45-123 mm/year and 123-224
mm/year class are statistically significant. In general, these rates correspond to semi-arid and dry sub-humid regions.
The high concentrations in these areas can be due to intensive agricultural activities.

Other explanatory variables such as rainfall or land cover/land use were not considered in the final model. Indeed,
notwithstanding a variable such as land cover/land use strongly influences observed logtransformed mean nitrate
concentration (Fig. 6), it is related to other variables such as population density.  Hence, to avoid multicollinearity in
the final model, the land cover/land use variable is no longer included in the final model.

The final multiple linear regression (MLR) model using the four variables yields an $R^2$ of 0.65, indicating that 65% of
the variation in observed logtransformed mean nitrate concentration at the pan African scale is explained by the model.
The result of the model is globally significant because the p-value =2.422e-10 at 95% of the significant level. The
observed versus predicted log transformed mean nitrate concentration is shown in Fig. 9 and indicates that the MLR fits
the data well. A probability plot of model residuals indicates that they follow a normal distribution (Fig. 10). We
performed the Shapiro-Wilk test as an additional check on the distribution of nitrate residuals. Because the probability
associated with the test statistic is larger than 0.05, we accept the null hypothesis that the residuals follow a normal
distribution. Despite the fact that a few points have higher Cook'D values compared to the rest of the observation, they
were kept in the MLR to represent the whole range of nitrate concentration data. In order to check the regressions
assumptions of homoscedasticity, a plot of the residuals of logtransformed mean nitrate versus the predicted
logtransformed mean values is illustrated in Fig.11. We observe that the majority of observations are in the range of -2
to 2 except for two outliers observed in the bottom left part of the graph. The residual standard error of the
logtransformed mean nitrate is 0.91116 (ln (mg/L)). We observe that the residuals decreases with increasing predicted
nitrate concentrations. The Breusch-Pagan test was used to assess heteroscedasticity in the model residuals (BP=24.2773
and p-value= 0.042). With a p-value of 0.042, we reject the null hypothesis that the variance of the residuals is constant
and infer that heteroscedasticity is indeed present. As a results, we may expect some bias in the MLR model.

Similarly to the logtransformed mean nitrate concentration modelling, we developed another model corresponding to
the logtransformed maximum nitrate concentration. This model yielded only an $R^2$= 0.42 for the maximum values. The
explanatory variables which influence the logtransformed maximum nitrate concentration in groundwater are: depth to
groundwater, soil media, topography, rainfall, climate class and type of region. For the logtransformed minimum
concentration, the absence of normal distribution assumptions did not allow to develop a MLR model.
**5 Discussion**
We present in this study a data based method to assess the vulnerability of groundwater systems for water quality
degradation. We used the logtransform of reported nitrate concentration as a proxy for groundwater vulnerability.   We
present a statistical model to explain this proxy in terms of generic data at the pan-African scale. In a previous study we
evaluated the groundwater vulnerability for pollution at the pan-African scale using the generic DRASTIC approach
(Ouedraogo et al., 2016). Yet, the uncalibrated DRASTIC model predictions are subjected to quite some uncertainty, in
particularly due to the subjectivity in assigning the generic DRASTIC model parameters.  In contrast to this previous
study we focus in this paper on nitrate pollution which is a parameter that is strongly related to vulnerability and that





often is measured in on-going monitoring programmes. We integrate published nitrate in groundwater data explicitly in
the assessment, thereby eliminating completely the subjectivity of the DRASTIC approach. The study also targets the
optimal use of the available data for the prediction purposes. Certain data might be redundant or biased. We addressed
in this study the quality of data (**Sect. 3.2**). In this study, we used multiple linear regression (MLR) for explaining nitrate
groundwater in terms of other generic spatially distributed environmental parameters. MLR is an approach to model the
relationship between a response variable and multiple set of explanatory variables (Rawlings et al., 1998). MLR analysis
is capable of both predicting and explaining a response variable using explanatory variables without compromise
(Kleinbaum et al., 1988). Previous studies of MLR using spatial variables for nitrate concentration in groundwater
showed $R^2$ values of 0.52 and 0.64 in shallow alluvial aquifers (Gardner and Vogel, 2005; Kaown et al., 2007) and $R^2$
of 0.82 in deep sandy tertiary aquifers (Mattern et al., 2009). For the application in this study, we selected the parameters
using stepwise MLR regression, allowing to select only those parameters which have significant impact on the
logtransformed concentration values of nitrate.

The explanatory variables with the strongest influence on the mean log transformed nitrate concentration at the pan
African scale are the population density and groundwater depth, which is in agreement with results from other studies
such as Nolan, (2001), Nolan et al., (2002), Nolan and Hitt (2006), Liu et al., (2013), Bonsor et al., (2011) and
Sorichetta et al. (2013). Both explanatory variables are directly related to the probability of having high nitrate
concentrations in groundwater. The strong influence of the population density variable can be explained by the serious
problem of sanitation in Africa townships. This is consistent with the conclusions of the UNEP/UNESCO project
'Assessment of Pollution Status and Vulnerability of Water Supply Aquifers Cities', stating that the major pollution
pressure on African water bodies are related to poor  on-site sanitation, solid waste dumpsites including household waste
pits and  surface water influences (Xu and Usher, 2006).  This is also consistent with other studies stating that that
leaking septic tanks and sewer systems are considerably causing nitrate contamination of groundwater in urban areas
(Bohlke, 2002; Showers et al., 2008). The magnitude of contamination is not only affected by the population density,
but also by the socio-economic setting (UNEP/DEWA, 2014). A high population density is therefore often associated
with the lack of adequate sanitation in many slums/shanty towns in Africa. The strong influence of population density
in our model suggests that high concentrations in groundwater are mainly from subsurface leakage of municipal sewage
systems, petrol service station (underground storage tanks), and agricultural chemicals in small scale farming. Hence,
sanitation programmes in Africa must not be delinked from groundwater protection and controlling the use of fertilizer
products in agriculture.

Nitrate concentrations were generally higher values for shallower wells than for deeper groundwater systems. The
inverse relation between depth and nitrate is consistent with previous groundwater studies that considered well depth or
depth of the screened interval as explanatory variables (Nolan and Hitt, 2006; Nolan et al., 2014; Wheeler et al., 2015;
Ouedraogo and Vanclooster, 2016). Deeper groundwater typically is older and may predate periods of intensive fertilizer
application (1950–present). Also, given the larger travel times associated with the recharge of deep groundwater
systems, there is an enhanced opportunity for denitrification (Wheeler et al., 2015). The same author showed that the
lagged increase in groundwater nitrate concentration relative to the increase of animal feeding operations suggests
groundwater recharge times of years or decades. Similar large recharge times for deep sandy aquifers were also
identified by Mattern and Vanclooster (2009).






The strong relation between nitrate contamination and both, groundwater depth and population density is a particular
point of concern given the fact that the majority (85 %) of Africa's population lives in regions where depth to
groundwater is shallow (0-50 m bgl) and where humps may be used to abstract water. Eight percent of these people (i.e.
nearly 66 million people) are likely to live in areas where depth to groundwater is 0-7 m bgl. A significant minority (8
%) of Africa's population lives in regions where the depth to groundwater is between 50 and 100 m bgl and common
hand pump technologies (e.g. India Mark) are inoperable in these cases.  These areas are mainly within southern Africa
and to a lesser extent situated in the Sahel.

A third important explanatory variable that was included in the model was the groundwater recharge rate. This is
consistent with studies like Hanson (2002) and Saffigna and Keeney (1997). According to UNEP/DEWA (2014),
recharge from multiple sources influences groundwater microbial and chemical water quality. Groundwater recharge
rate is interlinked with many other environmental variables including, but not limiting, soil type, aquifer type, antecedent
soil water content, land use / land cover type and rainfall (Sophocleous, 2004;  Ladekarl et al., 2005; Anuraga et al.,
2006). Hence, to avoid multi-collinearity, variables like land use/land cover type, rainfall, and soil type were not
considered in the final model.

Despite land cover/land use type is not explicitly included in the final model, the exploratory analysis clearly shows a
strong relationship between nitrate concentration and land use/land cover type. Indeed, nitrate concentrations are
generally higher in urban areas. This is consistent with many other studies such as Showers et al., (2008). The high
contamination in urban areas jeopardises groundwater exploitation in urban areas. Urbanization is a pervasive
phenomenon around the world, and groundwater demands in urban areas are increasingly growing. The degradation of
groundwater bodies in urban areas is therefore a particular point of concern. Also agricultural land exhibit an impact on
groundwater nitrate concentrations compared to the grassland/shrubland, water bodies and forest/bare area, but this
effect is less important as compared to agricultural land effects in other parts of the world (e.g. Europe).

The influence of aquifer type to the nitrate contamination was demonstrated by Boy-Roura et al.,(2013) and the
influence of soil type by Liu et al.,(2013). As with land cover/ land use type, these variables were not retained in the
final model to avoid collinearity with recharge.

The advantage of the MLR technique is that it can be easily implemented, and that model parameters can be easily
interpreted if the possible interaction between variables is ignored. However, MLR cannot represent well the many non-
linear dynamics that are associated to the contamination of groundwater systems. The violation of the homoscedasticity
hypothesis for instance, indicate that some bias will be present in our MLR model. Standard statistical models employed
in distribution modelling, such as MLR, work under the assumption of independence in the residuals and
homoscedasticity. When heteroscedasticity is present, residuals may be autocorrelated. This will lead to inflated
estimates in degrees of freedom, an underestimation of the residual variances and an overestimation of the significance
of effects (Legendre and Fortin, 1989; Legendre, 1993; Dale and Fortin, 2002; Keitt et al., 2002). This may show that
others variables should be included in the model or that the system may be highly non-linear.







We could avoid heteroscedasticity and improve the modelling performance by introducing non-linear regression
techniques (Prasad et al.,2006) or by introducing additional variables in the model. . Indeed, many studies showed that
non-linear statistical models of groundwater contamination outperform as compared to linear models (e.g. Pineros-
Garcet et al.,2006; Mattern et al., 2009; Oliveira et al.,2012 and Wheeler et al.,2015). To uncover nonlinear relationships
non-parametric data mining approaches provide obvious advantages (Olden et al., 2008; Wiens, 1989; Dungan et   al.,
2002). Machine learning provides a framework for identifying other explanatory variables, building accurate
predictions, and exploring other nonlinear mechanistic relationships in the system. We may therefore expect that non-
linear statistical models will improve the explanatory capacity of the model and remove heteroscedasticity from the
model.

However, we believe that this theoretical constraint of heteroscedasticity does not questions the overall results. The
observed heteroscedasticity can be considered modest in view of large extent of the study, and the violation of statistical
design criteria when collecting data through a meta-analysis. Also, the interpretation of the factors and coefficients
associated with non-linear regression techniques become more complicated. We therefore prefer to maintain in this
paper the MLR techniques as a first approach to screen the factors that contribute to logtransformed mean nitrate
concentration risk.  We suggest however that future studies should address the added value that can be generated with
non-linear modelling techniques.  Such non-linear modelling techniques are particularly needed for the maximum
concentration for which the $R^2$ of simple MLR remains currently too poor and also for the minimum concentration who
shows the absence of normal distribution assumptions.

Also, in this study, we only identified a MLR model based on a meta-analysis spanning the pan-African continent.
Since, the data collected through the meta-analysis are very heterogeneous, the quality of the data set remains rather
poor. Therefore, future studies should critically address the validity of the identified model and explore how the model
can be improved and be used in a predictive model.  It is however suggested that such model improvement and validation
step should be based on a more homogeneous data set. We therefore suggest to perform this future model validation
and model improvement step using data collected at the regional scale using more homogeneous data collection
protocols.

**6 Conclusion**
Contamination of groundwater by nitrate is an indicator of groundwater quality degradation and remains a point of
concern for groundwater development programmes all over the world. It is also a good proxy of overall groundwater
vulnerability for quality degradation.  We address in this paper the issue of nitrate contamination of groundwater at the
African scale. We inferred the spatial distribution of nitrate contamination of groundwater from a meta-analysis of
published field studies of groundwater contamination. We analysed the literature for reported mean, maximum and
minimum concentration of nitrate contamination.  We subsequently analysed, using box-plots, the reported





contamination in terms of spatially distributed environmental attributes related to pollution pressure and attenuation
capacity. We extracted the explanatory variables from a geographic information system with the ArcGIS 10.3™ tool.
We finally developed a MLR statistical model allowing to explain quantitatively the logtransformed observed
contamination that is a proxy of vulnerability in terms of spatially distributed attributes. We selected the explanatory
variables using a stepwise regression method.

We show that groundwater contamination by nitrates is reported throughout the continent, except for a large part of the
Sahara desert. The observed nitrate concentrations through meta-analysis ranges from 0 mg/L to 4625 mg/L for all
categories, i.e. mean, maximum and minimum values of nitrate groundwater contamination. The average mean nitrate
concentration is 27 mg/L. The distribution of the reported nitrate contamination data is strongly skewed. We therefore
build the statistical models for the logtransformed reported nitrate mean and maximum concentrations.

The graphical box plot analysis shows that nitrate contamination is important in shallow groundwater systems and
strongly influenced by population density and recharge rate. Nitrate contamination is therefore a particular point of
concern for groundwater systems in urban sectors.

The MLR model for the log transformed mean nitrate concentration uses the depth to groundwater, groundwater
recharge rate, and aquifer type and population density as explanatory variable. The total variability explained of the log
transformed reported mean nitrate concentration by this  analysis was 65 %, suggesting that other variables not
accounted for in the available ancillary data sets (such as climate zones) or better representations of the variable we do
considered may be needed to improve understanding of nitrate concentrations. These findings highlight the challenges
in developing appropriate regional variables to predict the conditions most vulnerable to high nitrate concentrations.
The MLR shows that the population density parameter is the most statistically significant variable. This authenticates
that leaking cesspits and sewer systems are considerably causing nitrate contamination of groundwater predominantly
in urban areas.  We identified similar MLR models for the log transformed maximum reported nitrate concentrations.
Yet, for this latter attribute, the explained variation using the simple MLR techniques (42 %) remains small.

One of the main strengths of our study is that it is based on a large database of groundwater contamination reports from
different countries, spanning the African continent and linked to environmental attributes that are available in a spatially
distributed high resolution format. In addition, the developing a continental-scale model of nitrate contamination in
groundwater of Africa, with its highly variable climate zones (hyperarid, arid, semi-arid, dry sub-humid, humid, tropical,
and Mediterranean)   allowed to determine which explanatory variables mainly influence the presence of nitrate. This
represents an important step in managing and protecting both water resources and human health, particularly in semi-
arid and arid regions. The main weakness or the major constraints of the modelling lies in the lack of detailed information
available at the pan African scale, particularly the lack and uneven distribution of measured nitrate points. In spite of
weaknesses and uncertainties caused by a moderate heteroscedasticity from residuals model, the modelling approach
presented here has great potential. Although the meta-analysis should not replace nitrate testing, it gives a first indication
of possible contamination; it can be also applied to preliminary assessment of nitrate using spatial variables and thus
may support the planning process and guidelines for transboundary aquifers managers and regional basin organizations.
This is particularly important as the demand for drinking water is increasing rapidly due to climate change and
population growth, which will undoubtedly increase the pressure on groundwater resources.



Finally, further development may include the use of non-linear modelling techniques such as Random Forest techniques
to identify the causal mechanism behind autocorrelation and heteroscedasticity in nitrate distributions over large extents
such as Africa. Such techniques have the potential to improve the quality of explanation and eventually prediction by
incorporate spatial autocorrelation, but complicate the physical explanation of observed trends. In addition, the model
should further be validated using more homogeneous data sets. There is a need for a process-based continental scale
nitrate estimate that uses a consistent approach and data, as the basis for studying potential environmental factors
impacts on groundwater resources in Africa. In a predictive mode, the model could be used for exposure estimate in
epidemiological studies on the effect of polluted groundwater on human health. Also, an application of the statistical
model to others contaminants could be explored.





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






**Table 1**. Criteria used to identify nitrate data studies within web data bases.

| Search engine | Search criteria |
|---|---|
| Google, Google Scholar, and Google Books | Groundwater pollution + Africa<br>Nitrate in groundwater + "African country name" or "African capital city name"<br><br>Groundwater quality + Africa<br><br>Nitrate and agricultural practices in Africa<br>Groundwater vulnerability + "African country name"<br><br>Pollution des eaux souterraines par les nitrates+ ''nom du pays Africain'' (in French)<br>Pollution des eaux souterraines + "nom du pays Africain" (in French)<br><br>Nitrate concentrations under irrigated agriculture + "African country name" |
| Web of Sciences, Scopus and Sciences Direct | Groundwater pollution by nitrate + "African country name"<br>Nitrate in groundwater +  "Africa capital city name"<br>Pollution des eaux souterraines par les nitrates + "nom du capital des pays"(in French)<br>Groundwater contamination by nitrate + "Africa countries" or  "African capital city name"<br>Africa irrigated agriculture + nitrate<br><br>Groundwater contamination by nitrate + "Africa country name" or  "African capital city name"<br>Nitrate concentrations under irrigated agriculture + "African country name"<br>Groundwater vulnerability to nitrate contamination + "Africa country name" or  "African capital city name" |
| Books | Groundwater pollution in Africa ( Xu and Usher, 2006) |






**Table 2.** Localisation of study sites considered in the meta-analysis

| Country | Localisation | Number of studies per country | References |
|---|---|---|---|
| Algeria | North east of Algeria | 11 | Labar et al.,2012a |
| | Ouargla phreatic aquifer in Algeria: Valley of OuedM'y a | | Semar et al., 2013 |
| | Nord-Algerian aquifer ( Mitidija) | | Sbargoud, 2013 |
| | Medja area | | Rouabhia et al., 2010 |
| | Biskra | | Messameh et al., 2014 |
| | Case Skikda | | Labar et al., 2012b |
| | El Eulma | | Belkhiri and Mouni, 2012 |
| | Mostaganem, Mecheria, Naama, Tiaret, Bechar, and Adrar | | Bahri and Saïbi, 2012 |
| | Southern Hodna | | Abdesselam et al., 2012 |
| | Tlemcen | | Abdelbaki et al., 2013 |
| | Merdja plain | | Rouabhia et al., 2009 |
| Angola | Angola | 1 | Angola Water Works, 2013 |
| Benin | Cotonou | 6 | Totin et al., 2013 |
| | Beninese coastal basin | | Totin et al., 2010 |
| | Municipality of Pobè | | Lagnika et al., 2014 |
| | Dongo-pont | | Bossa et al., 2012 |
| | Cotonou | | BGS,2003 |
| | Cotonou | | Xu et al.2006 |
| Bostwana | Rural Bostwana | 3 | Batisami, 2012 |
| | Kalahari | | Stadler et al., 2004 |
| | Eastern fringe of the Kalahari near Serowe | | Stadler et al., 2008 |
| Burkina Faso | Burkina Faso | 4 | BGS, 2002 |
| | Burkina Faso | | Pavelic et al., 2012 |
| | Sourou Valley | | Rosillon et al., 2012 |
| | Ouagadougou | | Xu et al.2006 |
| Cameroon | Mingoa River basin/Yaounde | 10 | Tabue et al., 2012 |
| | Bafoussam | | Mpakam et al.,2009 |
| | Coastal zone of Cameroon/Douala | | Nougang et al., 2011 |
| | Logon Valley/Chad–Cameroun | | Sorlini et al., 2013 |
| | Anga's river | | Kuitcha et al., 2013 |
| | Rio del Rey Basin/South Western Coast | | Wotany et al., 2013 |
| | Mingoa/Yaounde catchment | | Tabue et al., 2009 |


Hydrology and Earth System Sciences Discussions — Open Access — EGU

| Country | Location | # | References |
|---|---|---|---|
|  | 2 areas of Cameroon and Chad in the Lake Chad basin | | Ngatcha and Daira, 2010 |
|  | Dschang Municipality | | Temgoua, 2011 |
|  | Cameroon | | Xu et al.2006 |
| Central African Republic | Bangui area | 1 | Djebebe-Ndjiguim et al., 2013 |
| Congo-Brazzaville | Brazzaville | 5 | Matini et al., 2012 |
|  | Brazzaville | | Barhe and Bouaka, 2013 |
|  | Soult East Brazzaville | | Laurent et al., 2010 |
|  | South East Brazzaville | | Laurent and Marie, 2010 |
|  | South West Brazzaville | | Matini et al., 2009 |
| Egypt | Alexandria | 7 | Abd El-Salam and Abu-Zuid, 2014 |
|  | Helwan | | Abdalla and Scheytt, 2012 |
|  | Nile Valley | | Abdel-Lah and Shamrukh, 2001 |
|  | Tahta | | Easa and Abou-Rayan, 2010 |
|  | Kafr Al-Zayet District | | Masoud, 2013 |
|  | Nile Delta aquifers / Western Nile Delta | | Sharaky et al., 2007 |
|  | Cairo, Egypt / province of Giza | | Sadek and El-Samie, 2001 |
| Ethiopia | Dire Dawa | 13 | Abate, 2010 |
|  | Ethiopia | | BGS, 2001 |
|  | Raya valley | | Bushra, 2011 |
|  | Adis Ababa | | Engida, 2001 |
|  | Addis Ababa | | Kahssay et al. |
|  | Koraro/Tigray | | Nedaw, 2010 |
|  | Ethopia | | Pavelic et al., 2012 |
|  | Bulbule and Zway | | Bonetto et al., 2005 |
|  | Haromaya Watershed, Eastern Ethiopia | | Tadesse et al., 2010 |
|  | Akaki | | Tegegn, 2012 |
|  | Adis Ababa | | Xu et al.2006 |
|  | Dire Dawa of Sabian area | | Tilahun and Merkel, 2010 |
|  | Wondo Genet District, Southern Ethiopia | | Haylamicheal and Moges, 2012 |
| Ghana | Ga East | 14 | Ackah et al., 2011 |
|  | Sawla-Tuna-Kalba District | | Cobbina et al., 2012 |
|  | Akatsi, Adidome and Ho Districts | | Ansa- Asare et al., 2009 |
|  | Ghana | | BGS, 2000 |
|  | Six districts in the eastern region of Ghana | | Fianko et al., 2009 |
|  | Kwahu West District | | Nkansah et al., 2010 |
|  | Ga-East District of Accra( Taifa) | | Nyarko, 2008 |
|  | Ghana | | Obuobie and Barry, 2010 |
|  | Ghana | | Pavelic et al., 2012 |



| Country | Location | | References |
|---|---|---|---|
| | Densu basin | | Tay and Kortatsi, 2008 |
| | Contamination in Ghana | | Xu et al.2006 |
| | Western Region of Ghana | | Affum et al., 2015 |
| | Gold Mining area in Ghana/Tarkwa | | Armah et al., 2012 |
| | Lower Pra Basin of Ghana | | Armah, 2010 |
| Guinea-Biseau | Boloma | 1 | Bordalo and Savva-Bordalo, 2007 |
| | Bonoua | | Abenan et al., 2012 |
| | Bondoukou region | | Ahoussi et al., 2012 |
| | Bonoua aquifer (South-East Ivory Coast) | | Ake et al., 2010 |
| | Abidjan District | | Douagui et al., 2012 |
| | Adiaké Region | | Eblin et al., 2014 |
| | Abidjan and Korhogo | | Kouame et al., 2013 |
| | Abidjan aquifer | | Xu et al.2006 |
| | South-West Ivory Coast | | Yao et al., 2013 |
| | N'zi-Comoé (Centre East Ivory Coast) | | Kouassi et al., 2010 |
| | Guiglo-Douekoué (West Ivory Coast) | | Kouassi et al., 2012 |
| Ivory Coast | N'zi, N'Zianouan municipality(South Ivory Coast) | 16 | Ahoussi et al., 2012 |
| | Bandama basin at Tortiya(Nothern Ivory Coast) | | Drissa et al., 2013 |
| | Abia Koumassi village/Abidjan | | Loko et al., 2013a |
| | Slums of Anoumabo (Marcory) and AdjouffouPort-Bouet | | Osemwegie et al., 2013 |
| | Catchment Ehania, South-Eastern Ivory Coast | | Dibi et al., 2013 |
| | Hiré , South-West of Ivory Coast | | Loko et al., 2013b |
| Kenya | Kisaumi, Mombasa | 1 | Xu et al.2006 |
| | North-East Libya | | Nair et al., 2006 |
| Libya | Alshati | 3 | Salem and Alshergawi, 2013 |
| | North East Jabal Al Hasawnah | | Sanok et al., 2014 |
| | Lake Chilwa basin | | Xu et al.2006 |
| Malawi | Chikhwawa | 4 | Grimason et al., 2013 |
| | Upper Limphasa River/Nkhata-Bay district | | Kanyerere et al., 2012 |
| | Blantyre | | Mkandawire, 2008 |
| | Bamako city | | Xu et al.2006 |
| Mali | Mali | 3 | Pavelic et al., 2012 |
| | Timbuktu | | Cronin et al., 2007 |
| Mauritania | Mauritania | 1 | Friedel, 2008 |
| | Oued Taza | | Ben Abbou et al., 2014 |
| | Taldla plain | | Aghzar et al., 2002 |
| Morocco | Marrakesh | 16 | Alaoui et al., 2008 |
| | Meknès region | | Belghiti et al., 2013 |
| | Oum Azza of Rabat | | Benabbou et al., 2014 |





| Country | Count | Location | Reference |
|---|---|---|---|
|  |  | Phreatic aquifer of M'nasra | Bricha et al., 2007 |
|  |  | Taldla plain | EL Hammoumi et al., 2013 |
|  |  | Mzamza-Chouia | Asslouj et al., 2007 |
|  |  | Berrechid plain | EL Bouqdaoui et al., 2009 |
|  |  | Taldla plain | El Hammoumi et al., 2012 |
|  |  | Triffa plain | Fekkoul et al., 2011 |
|  |  | Triffa plain | Fetouani et al., 2008 |
|  |  | Essaouira Basin | Laftouhi et al., 2003 |
|  |  | Phreatic aquifer of Martil | Lamribah et al., 2013 |
|  |  | Casablanca | Smahi, 2013 |
|  |  | Souss-Massa basin (South-west Morocco) | Tagma et al., 2009 |
| Mozambique | 2 | Lichinga | Cronin et al., 2007 |
|  |  | Maputo city | Muiuane,2007 |
| Niger | 3 | Niamey | Chippaux et al., 2002 |
|  |  | Niamey | Hassane, 2010 |
|  |  | Niamey | Abou, 2000 |
| Nigeria | 24 | Uzouwani (South Eastern Nigeria) | Ekere, 2012 |
|  |  | Lagos | Adelekan and Ogunde, 2012 |
|  |  | Ondo State | Akinbile, 2012 |
|  |  | Southwestern Abeokuta | Aladejana and Talabi, 2013 |
|  |  | Lagos | Anthony, 2012 |
|  |  | Lagos-State | Balogun et al., 2012 |
|  |  | Nigeria | BGS, 2003 |
|  |  | Konduga town | Dammo et al., 2013 |
|  |  | Abuja | Dan-Hassan et al., 2012 |
|  |  | Nigeria | Edet et al., 2011 |
|  |  | Edo State/ South-South | Imoisi et al., 2012 |
|  |  | Jimeta-Yola (Northeastern of Nigeria) | Ishaku, 2011 |
|  |  | Eastern Niger Delta | Nwankwoala and Udom, 2011 |
|  |  | Anambra State | Obinna et al., 2014 |
|  |  | Lagos | Ojuri and Bankole, 2013 |
|  |  | Afikpo basin | Omoboriowo et al., 2012 |
|  |  | Benue State | Omguga, 2014 |
|  |  | Nigeria | Palevlic et al.,2012 |
|  |  | Niger Delta | Rim-Rukeh et al., 2007 |
|  |  | Igbokoda, Southwestern Nigeria | Talabi, 2012 |
|  |  | Lagos | Wakida and Lerner, 2005 |
|  |  | Nigeria | Xu et al.,2006 |
|  |  | Abia state | Obi and George, 2011 |




| Country | Location | | References |
|---|---|---|---|
| | Eti-Osa, Lagos | | Akoteyon and Soladoye, 2011 |
| Republic Democratic of Congo | Kahuzi-Biega Nationals Parks, Kinshasa | 2 | Bagalwa et al., 2013; Longo, 2009 |
| Senegal | Dakar Region; Thiaroye; Niayes region; Dakar; Dakar; Dakar; Yeumbeul/Dakar | 7 | Brandvold, 2013; Madioune et al., 2011; Sall and Vanclooster, 2009; Wakida and Lerner, 2005; Xu et al.,2006; Diédhiou et al., 2012; BGS, 2003 |
| Somalia | Somaliland and Puntland | 1 | FAO-SWALIM, 2012 |
| South Africa | South Africa; Philippi/Western Cape; Mpumalanga Province; South Africa; South Africa; Hex River Valley; Sandveld; Hertzogville | 6 | Maherry et al., 2009; Aza-Gnandji et al., 2013; Mpenyana-Monyatsi and Momba, 2012; Musekiwa and Majola, 2013; Pavelic et al., 2012; Xu et al.,2006 |
| Sudan | Southern Suburb of the Ondurman; Khartoun; Karrary; Karrary; Khartoum | 5 | Abdellah et al., 2013; Ahmed et al., 2000; Salim et al., 2014; Taha, 2010; Idriss et al., 2011 |
| Tanzania | Tanzania; Dar es Salam; Dodoma; Kilimandjaro region; Dar es Salam; Dar es Salam; Tanzania; Temekedistric/Dar es Salam | 8 | BGS, 2000; De Witte, 2012; Kashaigili, 2010; McKenzie et al., 2010; Mjemah, 2013; Mtoni et al., 2013; Palevlic et al.2012; Napacho and Manyele, 2010 |
| Tchad | N'djamena; Lake Chad basin; Chad basin | 3 | Guideal et al., 2010; Seeber et al., 2014; Ngatcha and Daira, 2010 |
| Togo | Agoè-Zongo; Gulf/South of Togo | 2 | Kissao and Housséni, 2012; Mande et al., 2012 |
| Tunisia | North-east of Tunisia (Korba aquifer); Cap Bon; Cap Bon; Djebeniana | 8 | Zghibi et al., 2013; Anane et al., 2014; Charfi et al., 2013; Fedrigoni et al., 2001 |



| | | | |
|---|---|---|---|
| | Metline-Ras Jebel-Raf Raf/North-East | | Hamza et al., 2007 |
| | Sfax-Agareb | | Hentati et al., 2011 |
| | El Khairat aquifer | | Ketata et al., 2011 |
| | Chaffar/ South of Sfax | | Smida et al., 2010 |
| Uganda | Uganda | | BGS, 2001 |
| | Kampala/Bwaise III | 2 | Kulabako et al., 2007 |
| | Petauke Town | | Mbewe, 2013 |
| | John Laing and Misisi de Lusaka | | Xu et al.2006 |
| Zambia | Copperbelt Province/(North Western Province; Lusaka Province ; Central Province ; Southern Province) | 4 | Nachiyunde et al., 2013 |
| | Lusaka | | Wakida and Lerner, 2005 |
| Zimbawe | Kamangara | 2 | Dzwairo et al., 2006 |
| | Epworth at Harare | | Zingoni et al., 2005 |



**Table 3.** Explanatory variables used in the MLR analysis.

| Explanatory variables | Type | Units or Categories | Spatial resolution/Scale | Date | Data source(s) |
|---|---|---|---|---|---|
| Land Cover/Land Use | Categorical data | - | 300 m | 2014 | [1]UCL/ELIe-Geomatics (Belgium) |
| Population density | Continuous point data | people/km2 | 2.5 km | 2004 | ESRI : www.arcgis.com/home |
| Nitrogen application | Continuous point data | kg/ha | 0.5° x 0.5° | 2009 | [2]SEDAC : www.sedac.ciesin.columbia.edu |
| Climate class data | Categorical data | - | 0.5° | 1997 | Global-Aridity values (UNEP, 1987)/ (UNESCO-IHE, Delft, The Netherlands) |
| Type of regions | Categorical data | - | 0.5° | 2014 | Global-Aridity values (UNEP, 1987)/ (UNESCO-IHE, Delft, The Netherlands) |
| Rainfall class | Categorical data | mm/year | 3.7 km | 1986 | UNEP : http://www.grid.unep.ch |
| Depth to groundwater | Categorical data | m | 0.5° x 0.5° | 2012 | British Geological Survey: www.bgs.ac.uk/ |
| Aquifer type | Categorical data | - | 1:3750 000 | 2012 | [3]GLiM data (Hamburg University) |
| Soil type | Categorical data | - | 1 km × 1 km | 2014 | ISRIC, World Soil Information: www.isric.org/content/soilgrids |
| Unsaturated zone(impact of vadose zone) | Categorical data | - | 1:3750 000 | 2012 | GLiM data (Hamburg University) |
| Topography/Slope | Continuous point data | Percentage (%) | 90 m | 2000 | UCL/ELIe-Geomatics (Belgium) and [4]CGIAR/CSI |
| Recharge | Continuous point data | mm/year | 5 km | 2008 | Global-scale modelling of groundwater recharge (University of Frankfurt) |
| Hydraulic conductivity | Continuous point data | m/day | Average size of polygon ~100km² | 2014 | [5]GLHYMPS data (McGill University) |

[1]Université Catholique de Louvain/Earth and Life Institute/Environnemental sciences;

[2]Socioeconomic Data and Applications Center (SEDAC);

[4]Consultative Group for International Agricultural Research (CGIAR)/ Consortium for Spatial Information (CSI);

[3]The new global lithological map database GLiM: A representative of rock properties at the Earth surface;

[5]A glimpse beneath earth's surface: Global Hydrogeology MaPS (GLHYMPS) of permeability and porosity.



**Table 4.** Summary statistics of original and log (ln) transformed nitrate data.

| Statistic | Maximum NO$_3^-$ concentration | Maximum ln(NO$_3^-$) concentration | Mean NO$_3^-$ concentration | Mean ln(NO$_3^-$) concentration | Minimum NO$_3^-$ concentration | Minimum ln(NO$_3^-$) concentration |
|---|---|---|---|---|---|---|
| Number of data (-) | 206 | 206 | 82 | 82 | 185 | 185 |
| Minimum (mg/l or ln(mg/l)) | 0.08 | -2.52 | 1.26 | 0.231 | 0 | 0 |
| Maximum (mg/l or ln(mg/l)) | 4625 | 8.43 | 648 | 6.473 | 180 | 5.19 |
| Median (mg/l or ln(mg/l)) | 73.64 | 4.29 | 27.58 | 3.317 | 0.55 | 0.43 |
| Mean (mg/l or ln(mg/l)) | 190.05 | 3.99 | 54.85 | 3.169 | 8.91 | 1.08 |
| Variance ((mg/l)$^2$ or ln(mg/l)$^2$) | 183778.94 | 3.39 | 163.92 | 43.901 | 537.07 | 1.78 |
| CV (-) | 225.56 | 46.18 | 8085.08 | 1.935 | 260.08 | 123.04 |
| Standard Deviation (mg/l or ln(mg/l)) | 428. 69 | 1.84 | 89.91 | 1.391 | 23.17 | 1.33 |
| Kurtosis | 60. 24 | 0.90 | 23.99 | -0.167 | 25.57 | 0.37 |
| Skewness | 6.75 | -0.74 | 4.31 | -0.294 | 4. 56 | 1.2 |





**Table 5.** Optimal linear regression model for explaining the logtransformed mean nitrate concentration

Coefficients:

|  | Estimate | Std. Error | t value | Pr (>|t|) |
|---|---|---|---|---|
| (Intercept) | 3.348e+00 | 6.624e-01 | 5.055 | 3.56e-06 *** |
| Depth [0-7] | 1.160e+00 | 3.895e-01 | 2.977 | 0.00404 ** |
| Depth [7-25] | 6.563e-01 | 3.693e-01 | 1.778 | 0.08002* |
| Depth [25-50] | 1.114e+00 | 4.755e-01 | 2.342 | 0.02216 ** |
| Depth [50-100] | 6.536e-01 | 4.005e-01 | 1.632 | 0.10744 |
| Depth [100-250] | 4.258e-01 | 6.766e-01 | 0.629 | 0.53129 |
| Recharge [0-45] | -2.506e-01 | 6.089e-01 | -0.412 | 0.68200 |
| Recharge [45-123] | -1.187e+00 | 6.055e-01 | -1.961 | 0.05407* |
| Recharge [123-224] | -1.112e+00 | 6.134e-01 | -1.812 | 0.07440* |
| Recharge [224-355] | -8.856e-01 | 6.089e-01 | -1.455 | 0.15047 |
| Aquifer media [Crystalline rocks] | -9.851e-01 | 3.374e-01 | -2.920 | 0.00477 ** |
| Aquifer media [Siliciclastic sedimentary rocks] | 1.893e-02 | 3.916e-01 | 0.048 | 0.96158 |
| Aquifer media [Unconsolidated sediments rocks] | -7.632e-01 | 3.384e-01 | -2.255 | 0.02740 ** |
| Aquifer media [Volcanic rocks] | -5.245e-01 | 6.123e-01 | -0.857 | 0.39469 |
| Population density (people/km$^2$) | 5.611e-04 | 6.887e-05 | 8.147 | 1.30e-11 *** |

Residual standard error: 0.9116 on 67 degrees of freedom
Multiple R-squared: 0.65
F-statistic: 8.693 on 14 and 67 DF, p-value=2.422e-10 < 0.001

Note: *** significant at p<0.001; ** significant at p<0.05 and * significant at p<0.1





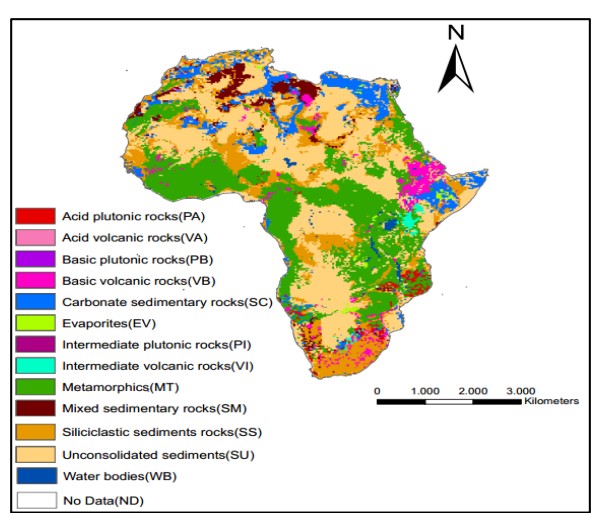

**Fig**. **1**. Hydrogeological setting of the African continent (from Hartmann and Moosdorf, 2012).

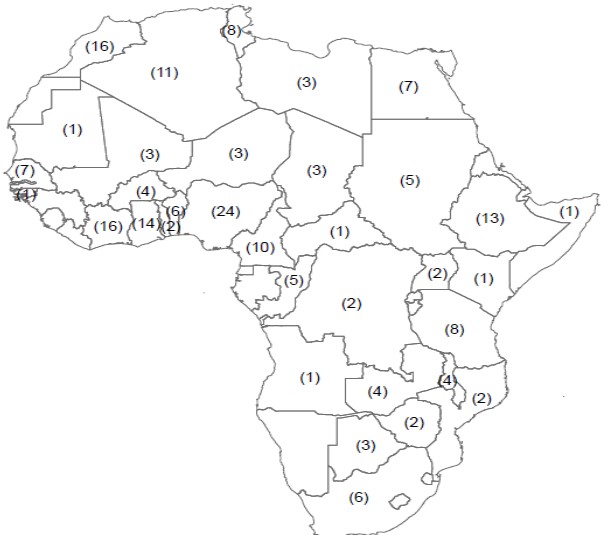

**Fig. 2.** Distribution of studies identified across Africa.





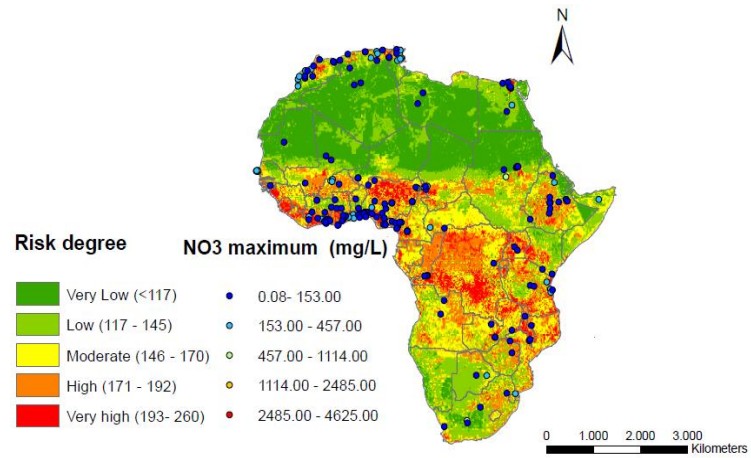

**Fig. 3.** The locations and the maximum values of nitrate in Africa superimposed on risk pollution map as generated in the previous generic vulnerability study of Ouedraogo et al., 2016.

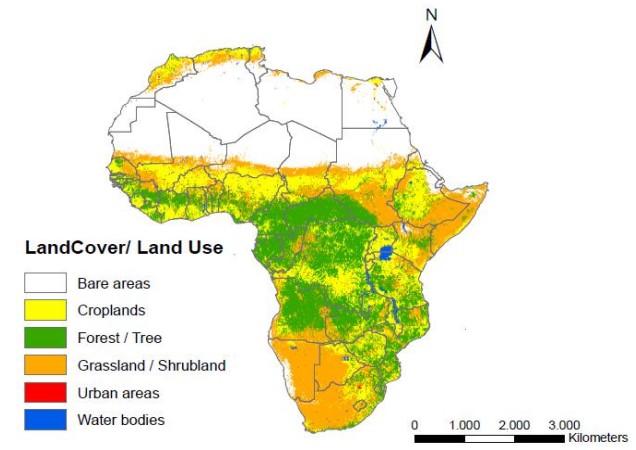

**Fig. 4**. Land Cover/Land Use map of Africa (modified from Defourny et al., 2014).




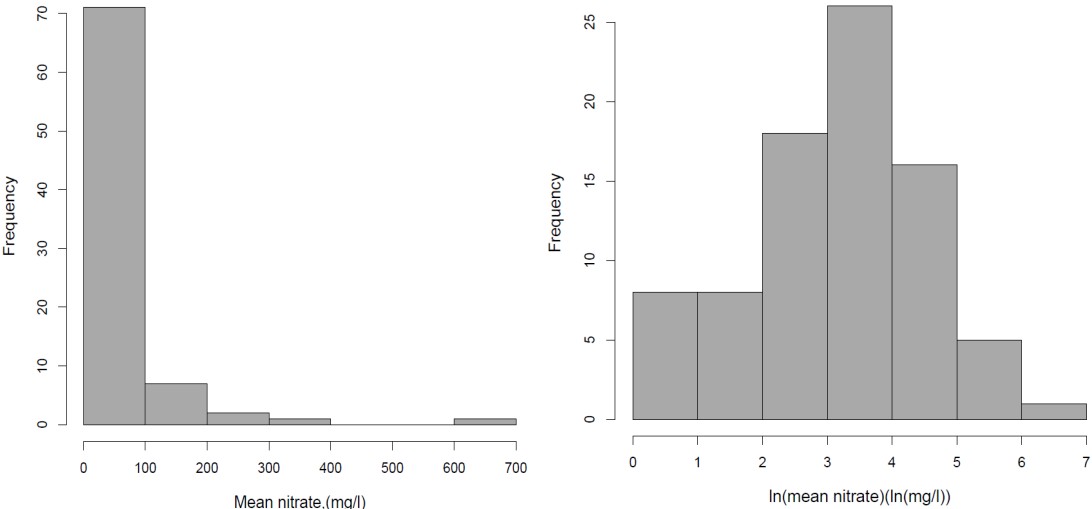

**Fig. 5**. Histograms of observed mean nitrate concentration (mg/l) and logtransformed mean nitrate concentration (ln (mg/l)).

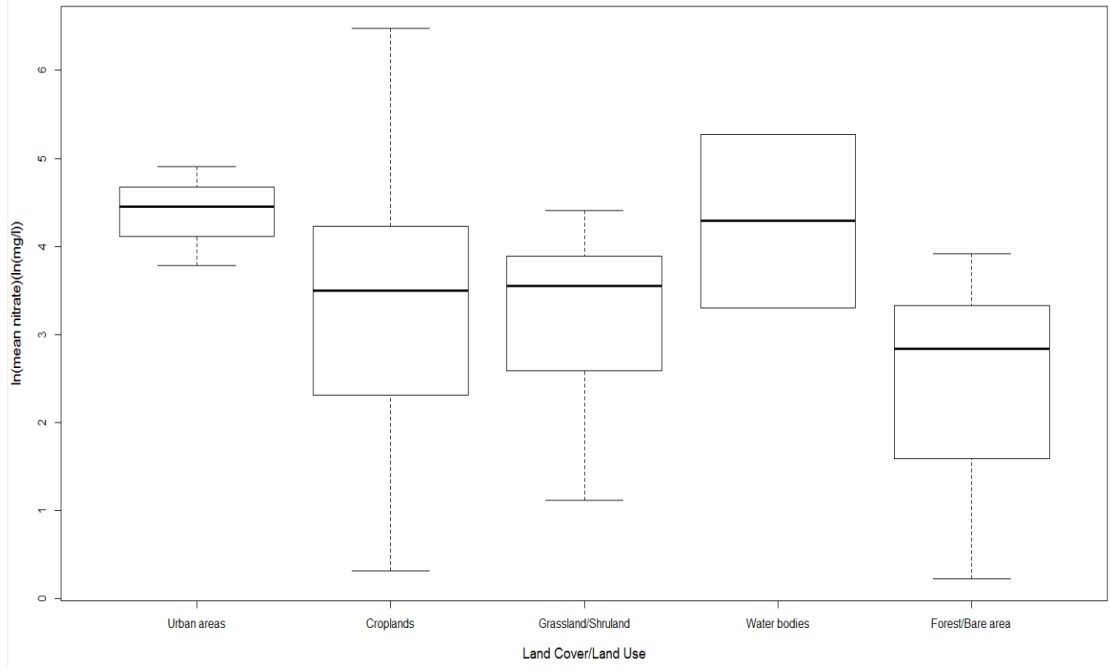

**Fig. 6**. Log transformed mean nitrate concentration for different land use classes.





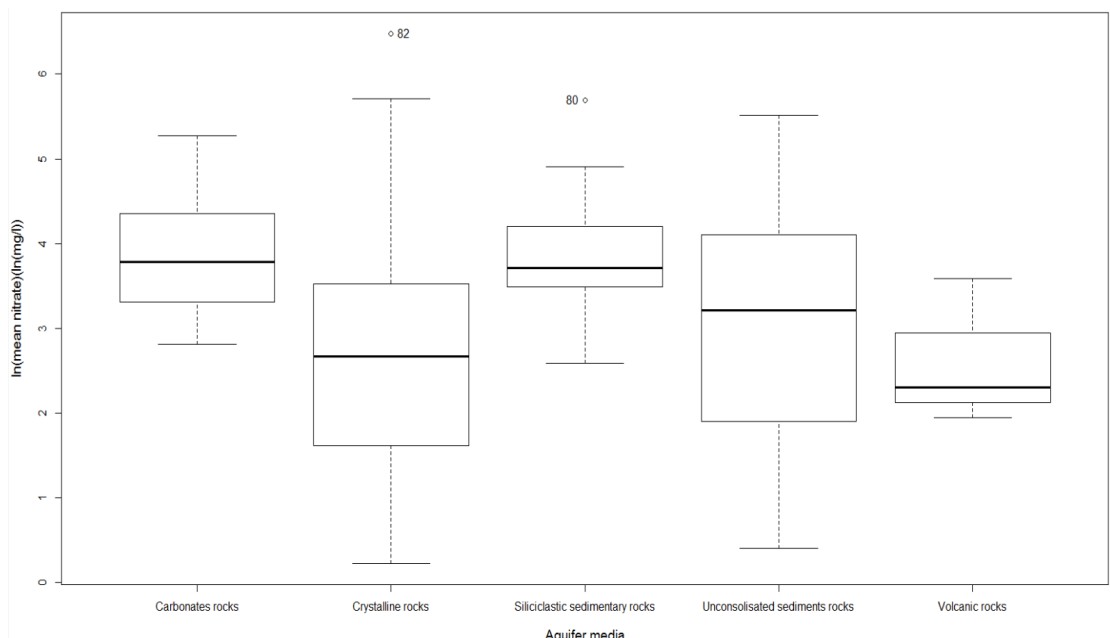

**Fig. 7.** Log transformed mean nitrate concentration for different aquifer system classes.





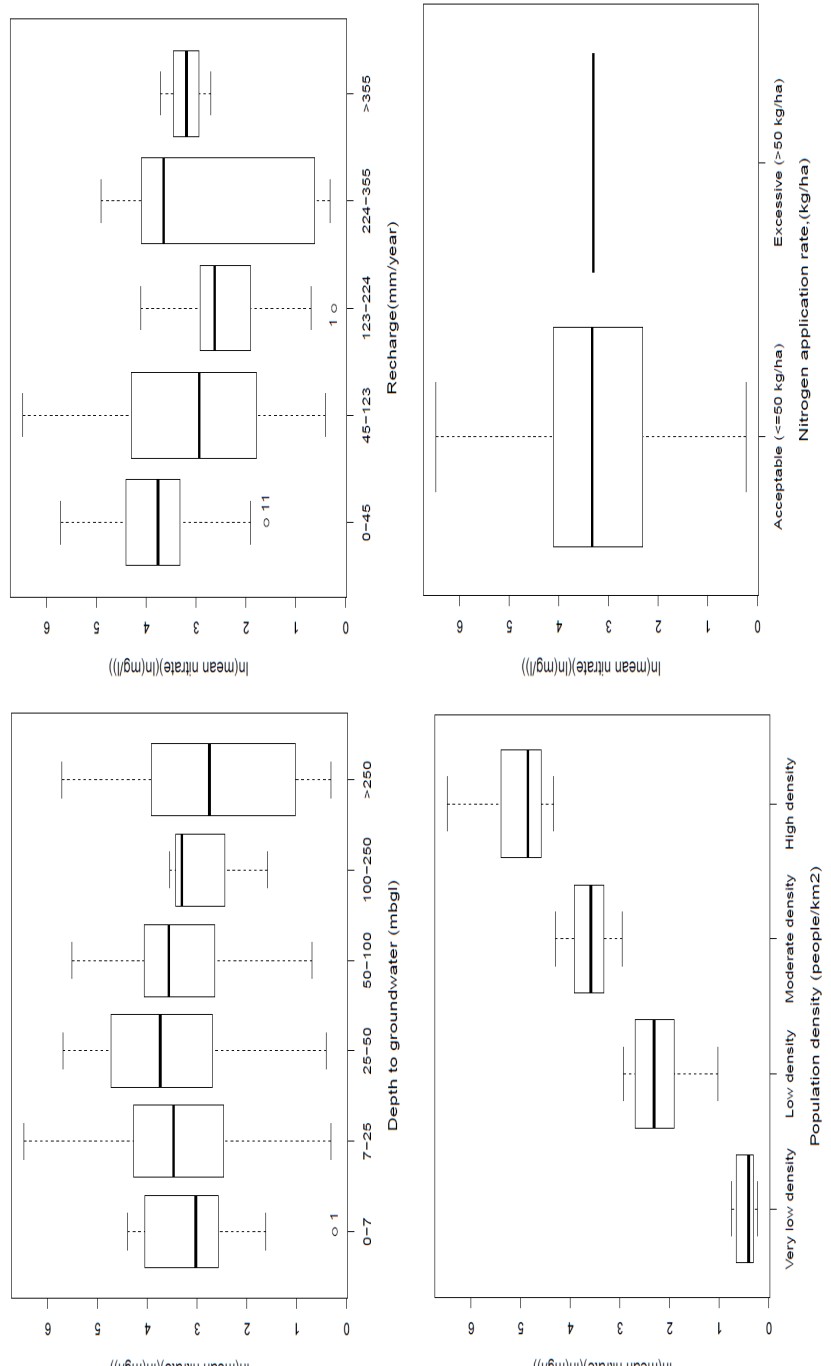

**Fig. 8.** Log transformed mean nitrate concentration for different groundwater depth classes, recharge classes, population density classes and nitrogen application rate classes.





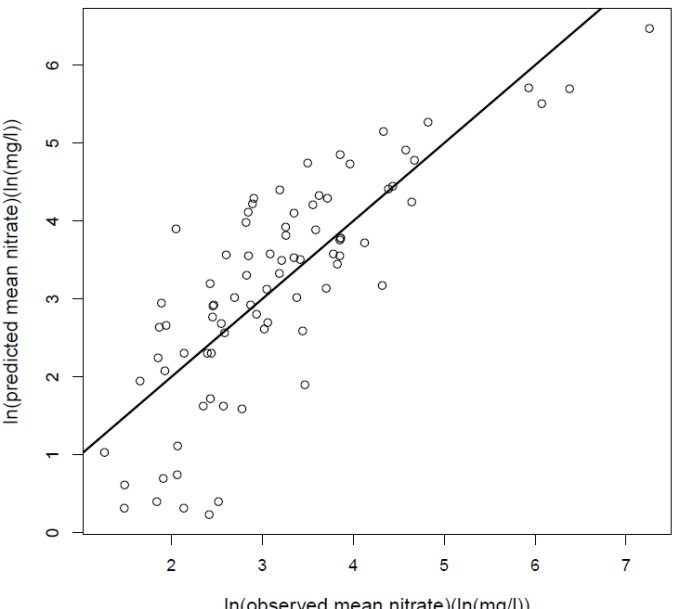

**Fig. 9.** Predicted versus observed mean logtransformed nitrate concentration (**R²=0.65**).

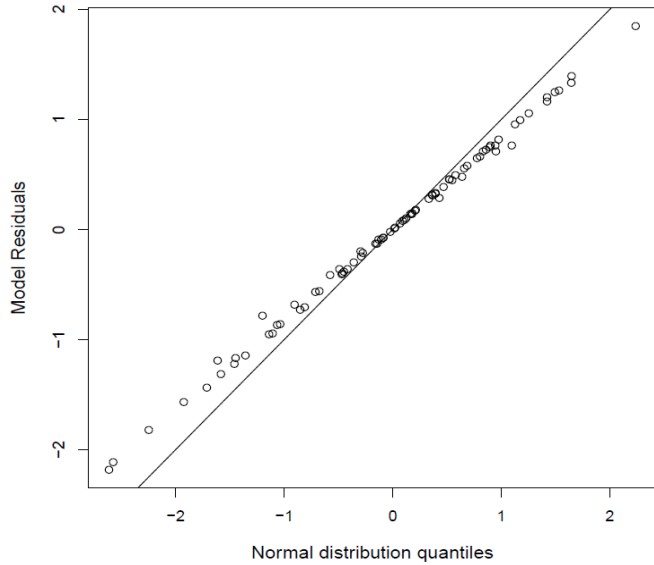

**Fig. 10.** Normal probability distribution of model residuals for the predicted logtransformed mean nitrate concentration.





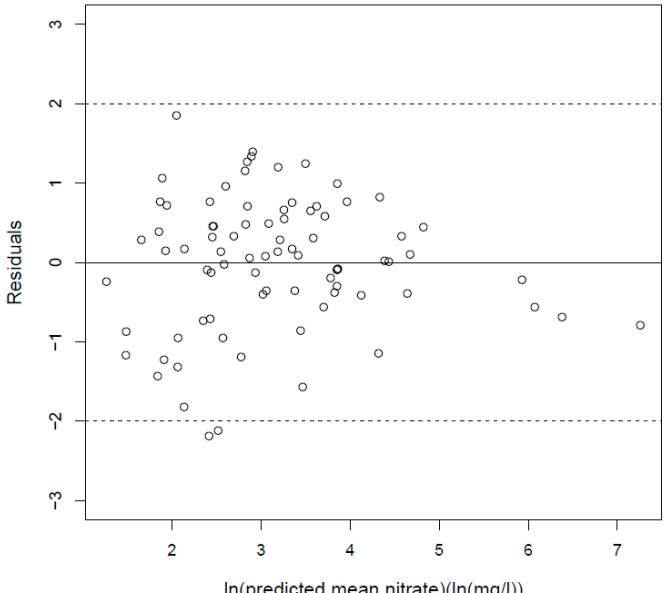

**Fig. 11.** Relation between residuals and predicted logtransformed mean nitrate concentration.