# Peer review of "A meta-analysis of groundwater contamination by nitrates at the"

_Hydrology and Earth System Sciences, 2016_

## Referee Comment (RC1) · Anonymous Referee #1 · 26 Apr 2016

General Comments

Ouedraougo and Vanclooster (2016) provide an interesting meta-analysis of groundwater nitrate studies across Africa. They build a simple multiple linear regression model to explore the potential factors affecting observed groundwater nitrate concentrations. The paper is interesting and within the scope of HESS. The novel contribution is the first synthesis of nitrate data across Africa. Unfortunately, the use of English through the paper is regularly strange and difficult to understand. I have included a few examples in the technical corrections but there are many more. It is suggested the paper is fully reviewed by a native English speaker prior to publication.

The issue of bias in the observed nitrate datasets analysed is not discussed in detail. It should be acknowledged explicitly in the text that a number of studies analysed have been investigating specific groundwater quality issues and thus the dataset may be biased towards higher concentrations. If possible, the number of studies that address specific groundwater nitrate contamination issues should be quantified against the number of more general groundwater hydrochemistry studies. I assume there will also be bias towards studies on aquifers which are productive and used for water supplies and this should also be noted.

A large number of datasets discussed in the paper are given as available "on request" from the author. Please provide these datasets as supplementary information.

Specific Comments

Title: I suggest "statistical modelling" is included in the title somewhere to better reflect the contents of the paper

P3L78 – I would be inclined to add a subheading in here for the section on methods for assessing groundwater vulnerability

Table 2 – It would be helpful to add another column showing for each study whether this is a peer reviewed journal article, book or other grey literature. As per the general comments, it would be helpful to detail for each study whether the study is addressing a nitrate contamination issue or is a more general hydrogeochemical study. This could be in the form of another column in the table.

P8L282-286 – I do not agree that increasing nitrate concentrations are observed with increasing recharge. If anything, they appear to decrease – is this due to dilution of nitrate in recharge?

P10 L357 – The probability plot in figure 10 shows a close to normal distribution but the points do not fall exactly on the straight line. The text should be changed to say that the distribution is close to normal.

L11 L414-422 The issues of lag in nitrate transport through the unsaturated zone and

denitrification are not well explained and should be re-written taking into account recent work on nitrate in the unsaturated zone. "Also, given the larger travel times associated with the recharge of deep groundwater systems, there is an enhanced opportunity for denitrification" – Do you mean there is more opportunity for denitrification in the unsaturated zone? Or in the saturated zones of deep confined aquifers? The evidence for unsaturated zone denitrification is limited (Kinniburgh et al., 1994; Rivett et al., 2008) and I do not think this should be used to explain why deeper aquifers have lower nitrate concentrations. There is a substantial body of literature showing evidence for nitrate accumulation in the unsaturated zone (Ascott et al., 2016; Wang et al., 2016; Worrall et al., 2015)– it may be that nitrate concentrations are lower in deeper aquifers because recharge from periods of high fertiliser use have not reached the water table yet.

Technical Corrections

P2L41 – remove "However,"

P2L59 – change "no comprehensive and synthetic study" to "no comprehensive synthesis of"

P2L63 – Define UN SDGs on first use

P2L67 – change "non-homogeneity" to "heterogeneity"

P9 L320 – do not need to explain how p-values work.

Subscript for 3 in NO3 not used consistently.

Figure 3 legend reports nitrate concentrations to 2 decimal places. This is not necessary – just report whole numbers.

P12 L426 – Humps – do you mean groundwater mounding?

Figure 8 looks stretched horizontally – please correct this. Please label each sub-figure (a), (b), (c) and (d) and refer to them in both the figure legend and text

References

Ascott, M.J., Wang, L., Stuart, M.E., Ward, R.S., Hart, A., 2016. Quantification of nitrate storage in the vadose (unsaturated) zone: a missing component of terrestrial N budgets. Hydrological Processes, DOI: 10.1002/hyp.10748

Kinniburgh, D.G., Gale, I.N., Gooddy, D.C., Darling W.G., Marks R.J., Gibbs B.R., Coleby L.M, Bird M.J., West J.M, 1994. Denitrification in the unsaturated zones of the British Chalk and Sherwood Sandstone aquifers. British Geological Survey, Keyworth, UK.

Rivett, M.O., Buss, S.R., Morgan, P., Smith, J.W., Bemment, C.D., 2008. Nitrate attenuation in groundwater: a review of biogeochemical controlling processes. Water Research 42, 4215-4232.

Wang, L., Stuart, M., Lewis, M., Ward, R., Skirvin, D., Naden, P., Collins, A., Ascott, M., 2016. The changing trend in nitrate concentrations in major aquifers due to historical nitrate loading from agricultural land across England and Wales from 1925 to 2150. Science of The Total Environment 542, 694-705.

Worrall, F., Howden, N.J.K., Burt, T.P., 2015. Evidence for nitrogen accumulation: the total nitrogen budget of the terrestrial biosphere of a lowland agricultural catchment. Biogeochemistry 123, 411-428.

---

## Referee Comment (RC2) · Anonymous Referee #2 · 29 Apr 2016

Overall, this is a thorough and well-thought-through evaluation of nitrate contamination of groundwater using a comprehensive modeling and literature review approach. I support its publication with minor revisions, largely focused on minor issues and lack of consistency in grammar use. I think of one of the key strengths of this manuscript is the solid and relatively rare linkage between the developed model and field-based (i.e., easily attainable in the field) data characterising NO3 pollution.

Itemised points:

-there are minor grammatical errors throughout; the manuscript is easily readable but not fully correct. Please have a native English speaker proof-read prior to final submission. - define all acronyms and use consistently. -The literature review is admirably

thorough, supporting the case. - The authors use fair and logical limitations on the data selected/used. - What does 'risk' describe in Fig. 3? How is 'risk' defined, per reader (and reviewer) understanding? - The authors have done an intensive analysis of the data provided via available literature. - Are nitrates naturally more abundant in specific geologic formations? If so, please include detail. - p. 14; the range of NO3 goes from 0 to 4625 mg/L for min and max.; this is a large range (as an aside, per the supporting text, max and min definitions need to be reversed). 1) std. dev. values should be included. 2) with this range of max and min, why is the avg. so low (27 mg/L)? The std. dev. for these data are needed to support. - There is a general lack of consistency in hyphen use throughout (e.g., Pan-African vs Pan African vs African). Please be consistent. - The importance of nitrate pollution is solidly presented. Can a brief discussion be included in the conclusion on how this issue can be addressed and /or alleviated?

---

## Author Comment (AC1) · 4 May 2016

HESS-2016-120, Reply by the authors to reviewer #1.

Dear Dr Editor,

We addressed all technical corrections for Anonymous Referee#1 as requested. We thank Anonymous Referee#1 for his positive and critical appreciation of our manuscript. Please find below a detailed revision report, along with the revised manuscript for your consideration. We have highlighted (yellow colour) the sections in the manuscript which have been amended or re-written.

**General Comments**

Ouedraogo and Vanclooster (2016) provide an interesting meta-analysis of groundwater nitrate studies across Africa. They build a simple multiple linear regression model to explore the potential factors affecting observed groundwater nitrate concentrations. The paper is interesting and within the scope of HESS. The novel contribution is the first synthesis of nitrate data across Africa. Unfortunately, the use of English through the paper is regularly strange and difficult to understand. I have included a few examples in the technical corrections but there are many more. It is suggested the paper is fully reviewed by a native English speaker prior to publication.

The issue of bias in the observed nitrate datasets analysed is not discussed in detail. It should be acknowledged explicitly in the text that a number of studies analysed have been investigating specific groundwater quality issues and thus the dataset may be biased towards higher concentrations. If possible, the number of studies that address specific groundwater nitrate contamination issues should be quantified against the number of more general groundwater hydrochemistry studies. I assume there will also be bias towards studies on aquifers which are productive and used for water supplies and this should also be noted.

**Response:**

We thank Anonymous Referee#1 for his constructive general comments. The English language was checked, but will further be improved through the HESS editing process (providing systematic language checking for all manuscripts). We agree with the Reviewer that bias in the dataset may be present. This issue was already addressed in the manuscript, but was not enough emphasized. We acknowledge that this bias may be due to multiples reasons, as stated by the reviewer. In our conclusion of the manuscript, we state that the main weakness or the major constraints of the modelling at the pan African scale, lies in the unavailability of a homogeneous data set on nitrate contamination, particularly the lack and uneven distribution of nitrate measurement points. Results from this analysis should not be over-interpreted. Whilst the data provide a useful preliminary assessment of the nitrate contamination in groundwater at the African scale, there are clear limitations. Unsurprisingly there are no consistent measurement datasets that can be explored at a continental scale; at larger scales, much of this information is also patchy, both spatially and temporally. Non-traditional data sources such as

literature data (for example meta-analysis) may therefore be useful substitutes for some traditional measurement data. The data used in this study are derived predominantly from literature. The data come from different sources (reviewed journal article, book of articles or other grey literature) and the methods (such as isotopic analysis) used to collected and produce the results of each study are not the same. They should therefore not be treated as traditional nitrate measurements.

We fully agree with the Reviewer that certain studies address specific groundwater nitrate contamination to water supply, others address groundwater nitrate contamination to strong irrigation areas, others again couple the two issues (water supply and irrigation), or again others studies address the nitrate pollution in mining zones, etc. The different nature of these studies constitute a possible bias. In spite of potential problems caused by possible sampling bias, the data set was used to explain the environmental/physical factors that contribute to the nitrate pollution at the African scale. Despite the issue of possible bias and uncertainties notified by the Reviewer, we are very optimistic about the robustness of the model for predicting contamination at the continental scale. The model that was obtained used population density, depth to shallow groundwater, aquifer type and recharge as explanatory variables. This is consistent with a study from UNEP/DEWA, (2014) in 11 countries across Africa that stated:

➢ "The level of protection at the wellhead strongly influences the quality of the well water. This is a vital aspect in protecting groundwater quality. Sanitation must not be delinked from Groundwater Protection".
➢ "Recharge from multiple sources influences groundwater microbial and chemical water quality".
➢ "The magnitude of contamination is also strongly affected by the population density and socio-economic setting".
➢ "Groundwater pollution and vulnerability issues are affecting all developing countries with increasing urbanization".

The study presented here is designed to give a continent-wide view of groundwater contamination by nitrates and to encourage the development of more qualitative national and sub-national qualitative model and assessments to support the development of groundwater-based adaptation strategies to current and future climate variability. Inevitably these results can be improved. They should be viewed as a first attempt to provide quantitative statistical modelling of nitrate contamination in groundwater for Africa and provides furthermore, a strong basis for future studies when homogeneous data without bias will be available at the African scale. Some elements of this discussion will be better emphasized in the revised manuscript (See P11 L391-402)

**: A large number of datasets discussed in the paper are given as available "on request" from the author. Please provide these datasets as supplementary information.**

**Response:** We thank for this suggestion. We will upload an excel file data set as supplementary information.

**Specific Comments**

**: Title: I suggest "statistical modelling" is included in the title somewhere to better reflect the contents of the paper**

**Response:** We thank Anonymous Referee#1 for this suggestion. The title was changed to incorporate the contents of the paper. The new title proposed is: "**A meta-analysis and statistical model of nitrates in groundwater at the African scale**". (See P1 L1).

**: P3L78 – I would be inclined to add a subheading in here for the section on methods for assessing groundwater vulnerability.**

**Response:** We thank the Reviewer for this suggestion. However, we suggest to keep this section in the introduction part. In HESS papers, the literature review is very often an integral part of the introduction. We prefer to keep this traditional structure.

**: Table 2 – It would be helpful to add another column showing for each study whether this is a peer reviewed journal article, book or other grey literature. As per the general comments, it would be helpful to detail for each study whether the study is addressing a nitrate contamination issue or is a more general hydro geochemical study. This could be in the form of another column in the table.**

**Response:** We thank the Reviewer for this suggestion. However, we think that adding an additional column decreases the readability and is redundant of the information the reader can retrieve in the reference list that is an integral part of our manuscript (Table 2 in P30-35).

**: P8L282-286 – I do not agree that increasing nitrate concentrations are observed with increasing recharge. If anything, they appear to decrease – is this due to dilution of nitrate in recharge?**

**Response:** We agree with the Reviewer: the nitrate decreases with recharge. This may be due to dilution effect into aquifers during recharge periods. We checked and corrected these errors in the manuscript. (See P8 L290-293).The dilution role for recharge factor was discussed in addition. (See also P13 L474-479).

**: P10 L357 – The probability plot in figure 10 shows a close to normal distribution but the points do not fall exactly on the straight line. The text should be changed to say that the distribution is close to normal.**

**Response:** We corrected the sentence to take account that the distribution is close to normal distribution. (See P10 L363).

**: L11 L414-422 The issues of lag in nitrate transport through the unsaturated zone and denitrification are not well explained and should be re-written taking into account recent work on nitrate in the unsaturated zone. "Also, given the larger travel times associated with the recharge of deep groundwater systems, there is an enhanced opportunity for denitrification" – Do you mean there is more opportunity for denitrification in the unsaturated zone? Or in the saturated zones of deep confined aquifers? The evidence for unsaturated zone denitrification is limited (Kinniburgh et al., 1994; Rivett et al., 2008) and I do not think this should be used to explain why deeper aquifers have lower nitrate concentrations. There is a substantial body of literature showing evidence for nitrate accumulation in the unsaturated zone (Ascott et al., 2016; Wang et al., 2016; Worrall et al., 2015)– it may be that nitrate concentrations are lower in deeper aquifers because recharge from periods of high fertiliser use have not reached the water table yet.**

**Response:** We agree with these remarks on the denitrification process. We rewrote the L430-457 in P 12 to include recent work on this issue.

Indeed, according to Close (2010), nitrate is negatively charged and thus electrostatically repelled by media in unsaturated zone that usually have a negative charge, such as clay minerals. This means that nitrate is less likely to be sorbed within the unsaturated zone. Moreover, according to Rivett et al., (2008), denitrification, which is generally facilitated by the absence of oxygen, is considered to be the dominant nitrate attenuation process in the subsurface system. Denitrification was found to be relatively limited in unsaturated zone (Kinniburgh et al., 1994; Rivett et al., 2008), while it is the principle process responsible for reduction of nitrate in groundwater (Aljazzar, 2010, Stevenson and Cole, 1999; Thayalakumaran et al., 2004), in particular in reduced groundwater (Burow et al.,2013). Boy-Roura et al.,(2013), for instance found low nitrate concentrations (below 50 mg/L) in those areas where denitrification processes have been identified. An indicator of the presence of denitrification processes contributed as such to explain nitrate contamination in the Osona region (NE Spain) (Boy-Roura et al., 2013). In our study, an indicator of the presence of denitrification processes in the groundwater system was not available and could not be included in the model.

**Technical Corrections**

**: P2L41 – remove "However,"**

**Response:** The word "However" was removed in the revision manuscript. (See P2 L42).

**: P2L59 – change "no comprehensive and synthetic study" to "no comprehensive synthesis of"**

**Response:** We changed "no comprehensive and synthetic study" to "no comprehensive synthesis of". (See P2 L60-61).

**: P2L63 – Define UN SDGs on first use**

**Response:** UN SDGs is the United Nations (UN) Sustainable Development Goals (SDGs). Furthermore, according UN, (2014), transboundary water cooperation is a focus of target 6.5, which states "by 2030 implement integrated water resources management at all levels, including through transboundary cooperation as appropriate". According to Saruchera and Lautze, (2015), transboundary water cooperation has emerged as an important issue in the post-2015 United Nations (UN) Sustainable Development Goals (SDGs). (See P2 L63-67).

**: P2L67 – change "non-homogeneity" to "heterogeneity"**

**Response:** We substituted "non-homogeneity" by "heterogeneity" term. (See P2 L70).

**: P9 L320 – do not need to explain how p-values work.**

**Response:** We eliminated the sentence that explains how p-values work in the manuscript. (See P9 L326).

**: Subscript for 3 in NO3 not used consistently.**

**Response:** We checked it by replacing by $NO_3$ (see P2 L35).

**: Figure 3 legend reports nitrate concentrations to 2 decimal places. This is not necessary – just report whole numbers.**

**Response:** We built a new figure without any decimal. We retained just whole numbers (see figure 3, P40 or figure in below).

[Figure]

Figure 3 in the manuscript.

**: P12 L426 – Humps – do you mean groundwater mounding?**

**Response:** This error was replaced by "hand pumps". (See P13 L468).

**: Figure 8 looks stretched horizontally – please correct this. Please label each sub-figure (a), (b), (c) and (d) and refer to them in both the figure legend and text**

**Response:** We corrected and used the labels (a), (b), (c) and (d) for respectively groundwater depth classes, recharge classes, population density classes and nitrogen application rate classes. This figure was also adjusted and centred to eliminate the horizontal stretching. (See figure 8 in the manuscript, P43).
Moreover, in the text, we use the label for each figure. (See P8, L283, L290, L296 and L304).

**References cited**

Aljazzar, T. H.: Adjustment of DRASTIC Vulnerability Index to Assess Groundwater Vulnerability for Nitrate Pollution Using the Advection Diffusion Cell. Von der

Fakultät für Georessourcen und Materialtechnik der Rheinisch-Westfälischen Technischen Hochschule Aachen, 146pp., 2010.

Boy-Roura, M., Nolan, B. T., Mencló, A. and Mas-Pla, J.: Regression model for aquifer vulnerability assessment of nitrate pollution in the Osona region (NE Spain). Journal of Hydrology 505: 150-162., 2013

Boy Roura, M: Nitrate groundwater pollution and aquifers vulnerability: the case of the Osona region. Ph.D thesis. Universitat de Girona, 143 pp., 2013.

Burow, K. R., Jurgens, B. C., Belitz, K., & Dubrovsky, N. M: Assessment of regional change in nitrate concentrations in groundwater in the Central Valley, California, USA, 1950s–2000s. Environmental earth sciences,69(8), 2609-2621., 2013.

Close,M.: Critical Review of Contaminant Transport Time Through the Vadose Zone. Environment Canterbury Technical Report. Report No. R10/113 ISBN 978-1-927137-54-3 (46 p.).,2010 (accessed online 28 April 2016).

Nolan, B. T., Gronberg, J. M., Faunt, C. C., Eberts, S. M., & Belitz, K.: Modeling nitrate at domestic and public-supply well depths in the Central Valley, California. Environmental science & technology, 48(10), 5643-5651., 2014.

Rivett, M. O., Buss, S. R., Morgan, P., Smith, J.W.N.,Bemment, C.D.: Nitrate attenuation in groundwater: a review of biogeochemical controlling processes. Water Res 42(16): 4215-4232., 2008.

Saruchera, D. and J. Lautze, J: Measuring Transboundary Water Cooperation: Learning from the Past to Inform the Sustainable Development Goals, IWMI. Working Paper 168. 20pp., 2015.

Stevenson, F.J., and Cole, M.A., (1999): Cycles of Soil Carbon, Nitrogen, Phosphorus, Sulfur, Micoronutrients (2nd ed.), John Wiley and Sons Inc.

Thayalakumaran, T., Charlesworth, P.B., Bristow, K.L., van Bemmelen, R.J., Jaffres, J.: Nitrate and Ferrous Iron Concentration in the Lower Burdekin Aquifers: Assessing Denitrification Potential. SuperSoil 2004: 3rd Australian New Zealand Soils Conference, 5 – 9 December 2004, University of Sydney, Australia (accessed online 27 April, 2016).

UN (United Nations). 2014. Report of the open working group on Sustainable Development Goals. Available at http:// sustainabledevelopment.un.org/focussdgs.html (accessed on April 27, 2016).

UNEP/DEWA.: Sanitation and Groundwater Protection –a UNEP Perspective UNEP/DEWA, http://www.bgr.bund.de/EN/Themen/Wasser/Veranstaltungen/symp_sanitatgwprotect/present_mmayi_pdf.pdf?__blob=publicationFile&v=2: 18pp.,(last access: 13 August 2014).

---

## Author Comment (AC2) · 4 May 2016

HESS-2016-120 Revision report

"A meta-analysis of groundwater contamination by nitrates at the African scale" by Issoufou Ouedraogo and Marnik Vanclooster,(2016)

Dear Dr Editor,

We addressed all technical corrections for Anonymous Referee#2 as requested. We thank Anonymous Referee#2 for his positive and critical appreciation of our manuscript. Dear Anonymous Referee#2, please find below a point-by-point revision report, along with the revised manuscript for your consideration. We have highlighted (blue colour) the sections in the manuscript which have been amended or re-written.

Overall, this is a thorough and well-thought-through evaluation of nitrate contamination of groundwater using a comprehensive modeling and literature review approach. I support its publication with minor revisions, largely focused on minor issues and lack of consistency in grammar use. I think of one of the key strengths of this manuscript is the solid and relatively rare linkage between the developed model and field-based (i.e., easily attainable in the field) data characterising NO3 pollution.

Itemised points:

-there are minor grammatical errors throughout; the manuscript is easily readable but not fully correct. Please have a native English speaker proof-read prior to final submission.

**Response:** We thank Anonymous Referee#1 for his constructive general comments. The English language was checked, but will further be improved through the HESS editing process (providing systematic language checking for all manuscripts).

- define all acronyms and use consistently.

**Response:** We present a list of acronyms at the end of the manuscript (see P16 L590-601):

**BGS**:         British Geological Survey
**DEWA**:        Division of Early Warning and Assessment
**DRASTIC**:     Depth, Recharge, Aquifer media, Soil media, Topography, Impact of vadose zone, Conductivity
**FAO**:         Food and Agriculture Organization (United Nations)
**OECD**:        Organization for Economic Cooperation and Development
**MODFLOW**:     MOdular finite-Difference Flow model (U.S. Geological Survey)
**UNEP**:        United Nations Environment Programme
**UNESCO**:      United Nations Educational, Scientific and Cultural Organization
**US EPA**:      United States Environmental Protection Agency

-The literature review is admirably thorough, supporting the case. - The authors use fair and logical limitations on the data selected/used. - What does 'risk' describe in Fig. 3? How is 'risk' defined, per reader (and reviewer) understanding?

**Response:** We defined the "risk" in the text as follows (see P5 L173-178).

The groundwater pollution risk corresponds to the potential of a groundwater body for undergoing groundwater contamination (Farjad et al., 2012). The risk of pollution is determined both by the intrinsic vulnerability of the aquifer, which is relatively static, and the existence of potentially polluting activities at the soil surface. These latter activities are time dynamic and can be controlled (Saidi et al, 2010). We generated the groundwater pollution risk map by combining the intrinsic groundwater vulnerability map with the land use map, using the additive model of Secunda et al. (1998). Details of these procedure are given by Ouedraogo et al. (2016).

– The authors have done an intensive analysis of the data provided via available literature. - Are nitrates naturally more abundant in specific geologic formations? If so, please include detail.

**Response:** We addressed this remark in P12-13, L458-464.
"Geologic nitrogen" was first recognized by Boyce et al., (1976) as nitrogen associated with certain formations, sedimentary and inorganic in origin. Holloway and al., (1998), investigated the contribution of bedrock nitrogen to high nitrate concentrations in stream water. They conclude that the weathering of nitrogen from rock can potentially affect the chemistry of water and soil. We cited in the revised manuscript the main conclusions of the study of Tredoux and Talma, cited in Xu and Usher, (2006):

➢ In most cases, the occurrence of high levels of nitrate is due to contamination related to anthropogenic activities.
➢ Geological formations can only serve as a primary source of nitrogen in exceptional cases where contamination ions are incorporated in rock minerals to be released by weathering and oxidized to nitrate.
➢ The apparent correlation between the occurrence of high nitrate levels and certain geological formations is due to secondary characteristics of the geological formation and associated factors allowing enrichment with nitrate derived from others sources.

The authors affirms that nitrate originating from anthropogenic sources is the major problem. In this regards, we can conclude that nitrate in specific geologic formations has a relative weak contribution to nitrate in groundwater, but, rather anthropogenic activities. We can classify the question of the Reviewer#2 in the same category of unsaturated zone denitrification process and nitrate accumulation in the unsaturated zone asked by the Reviewer#1. We addressed to this comment to the reply for Reviewer#1.

- p. 14; the range of NO3 goes from 0 to 4625 mg/L for min and max.; this is a large range (as an aside, per the supporting text, max and min definitions need to be reversed). 1) std. dev. Values should be included. 2) with this range of max and min, why is the avg. so low (27 mg/L)? The std. dev. for these data are needed to support.

Response: Maximum and minimum in the text in page 14 were reversed. 1) The standard deviation and the range of minimum and maximum were included in the text. The value 27 mg/L is the median for the mean nitrate concentration category. This was an error in the initial manuscript. We checked and correct this. We added the standard deviations for all data in the text (See P15, L550-555).

"The mean nitrate concentration varies between 1.26 to 648 mg/L. The sample mean of this mean nitrate concentration was 54.85 mg/L, its standard deviation was 89.91 mg/L and its median was 27.58 mg/L. The minimum nitrate concentration varies between 0 to 185 mg/L, while the maximum concentration varies 0.08 to 4625 mg/L), the sample mean of the minimum and maximum concentrations were 8.91 mg/l and 190.05 mg/L, while the sample standard deviations were 23.17 mg/L and 428.69 mg/L and the medians were 0.55 mg/L and 73.64 mg/L respectively."

- There is a general lack of consistency in hyphen use throughout (e.g., Pan-African vs Pan African vs African). Please be consistent.

Response: We checked and we eliminated all terms of "Pan-African", "Pan African". We use "the African scale" only in the revision manuscript.

- The importance of nitrate pollution is solidly presented. Can a brief discussion be included in the conclusion on how this issue can be addressed and/or alleviated?

Response:

The range of literature papers presented in our study illustrates the pollution problem at the continental scale. Our paper does not have the ambition to discuss the remediation problem. In order to keep the focus (the paper is already very lengthy), we propose not to include a discussion on the nitrate remediation problems in our manuscript. However, for the sake of the reviewer, we give some ideas below.

Groundwater protection and alleviation at the pan African scale is not optional. Remediation should be developed at the regional scale. The solutions that can be proposed to mitigate and improve the situation were already partially addressed by Xu and Usher, (2006):

i.    Political will: Because, groundwater quality protection is closely related to the government policy towards economic development and the political will for sustainable development and utilisation of resources. Our study may increase awareness of AMCOW (African Ministerial Council on Water) to proceed with groundwater protection programs at the Pan African level.

ii. Capacity building and technical skills: Africa, has little capacity to challenge groundwater degradation and there is a need to boost this capacity through appropriate capacity building programs.

iii. Knowledge dissemination: Awareness of groundwater resources in Africa is low. There is a need to improve the knowledge on groundwater systems for decision makers and for the broader public. Our paper may contribute to the increase of groundwater awareness.

In addition to these 3 points in above, we think that African decision-makers must elaborate groundwater protection programs that are based on groundwater monitoring and data management. Such programs can be boosted through multilateral organisation such as the African Groundwater Commission or SADC, ECOWAS, the Nubian Aquifer Regional Information Systems (NARIS), The North Western Sahara Aquifer System (NWSAS) (better known under the acronym SASS for its French name "Système Aquifère du Sahara Septentrional").

**References cited**

Boyce, J.S., Muir,J., Edwards, A.P., Seim, E.C., and Olson, R.A., Geologic nitrogen in Pleistocene loess of Nebraska. Journal of Environmental Quality. Vol. 5 No. 1, P. 93-96., 1976.

Farjad, B., Mohamed, T., A., Wijesekara N., Pirasteh, S., Shafri, H. Z. b. M.: Groundwater intrinsic vulnerability and risk mapping. Proceedings of the ICE - Water Management, 165(8), 441-450., 2012.

Holloway, J.M., Dahlgren, R.A., Hansen, B., and Casey, W.H.: Contribution of bedrock nitrogen to high nitrate concentrations in stream water. Nature, Vol.395. P.785-788., 1998.

Ouedraogo, I; Defourny, P; Vanclooster, M.: Mapping the groundwater vulnerability for pollution at the pan African scale. Science of the Total Environment, 544, 939–953,(December 2015).,2016.

Saidi, S., Bouri, S. and Ben Dhia, H.: Groundwater vulnerability and risk mapping of the Hajeb jelma aquifer (Central Tunisia) using a GIS-based DRASTIC model. Environmental Earth Sciences, 59(7), 1579-1588., 2010.

Secunda, S., Collin, M., &Melloul, A. J.:Groundwater vulnerability assessment using a composite model combining DRASTIC with extensive land use in Israel's Sharon region. Journal of Environmental Management, 54, 39–57., 1998.

Xu, Y. and Usher, B.: Groundwater pollution in Africa, Taylor&Francis/Balkema, The Netherlands. 353pp., 2006.

---

## Author Comment (AC3) · 4 May 2016

The comment was uploaded in the form of a supplement:
http://www.hydrol-earth-syst-sci-discuss.net/hess-2016-120/hess-2016-120-AC3-supplement.pdf

---

## Author Response (AR1)

HESS-2016-120, Reply by the authors to Editor comments

Dear Prof. Thomas Kjeldsen as handling Editor,

We replied to all comments that were formulated by the reviewers. We uploaded detailed revision reports on the HESS server.

With respect to the language issue, we confirm that this manuscript was thoroughly checked by both co-authors, and further polished with language correction software (Ms-Word language check / 'grammarly' add-in plug). We also rely on the English copy-editing service that is offered by the HESS editing office to further polish the language, if needed. We hope that with this, the English level will be sufficient for accepting publication in HESS.

Best regards.

Issoufou Ouedraogo